# COST-EFFECTIVE AGENT TEST-TIME SCALING VIA BUDGET-AWARE THINKING

## ABSTRACT

Scaling test-time computation improves performance across different tasks on large language models (LLMs), which has also been extended to tool-augmented agents. For these agents, scaling involves not only "thinking" in tokens but also "acting" via tool calls. The number of tool calls directly bounds the agent's interaction with the external environment. However, we find that simply granting agents a larger tool-call budget fails to improve performance, as they lack "budget awareness" and quickly hit a performance ceiling. To address this, we study how to scale such agents effectively under explicit tool-call budgets, focusing on web search agents. We first introduce the **Budget Tracker**, a lightweight plug-in that provides the agent with continuous budget awareness, enabling simple yet effective scaling. We further develop **BATS** (**B**udget **A**ware **T**est-time **S**caling), an advanced framework that leverages this awareness to dynamically adapt its planning and verification strategy, deciding whether to "dig deeper" on a promising lead or "pivot" to new paths based on remaining resources. To analyze cost-performance scaling in a controlled manner, we formalize a unified cost metric that jointly accounts for token and tool consumption. We provide the first systematic study on budget-constrained agents, showing that budget-aware methods produce more favorable scaling curves and push the cost-performance Pareto frontier. Our work offers empirical insights toward a more transparent and principled understanding of scaling in tool-augmented agents.

## 1 INTRODUCTION

Scaling test-time compute in large language models (LLMs) helps improve the performance across a wide range of tasks including reasoning, coding (Wu et al., 2025b; Snell et al., 2024; Muennighoff et al., 2025; Chen et al., 2025b). Mainstream scaling strategies such as sequential (Madaan et al., 2023) and parallel scaling (Wang et al., 2023; Brown et al., 2024) enable models to utilize more effort, elicit deeper reflection, and refine their outputs, often leading to substantial gains in answer quality (Zhang et al., 2025a). These successes motivate recent efforts to extend test-time scaling to tool-augmented agents (Zhu et al., 2025b; Wang et al., 2025a), where LLMs are equipped with various tools to interact with the external environment such as search engines or APIs.

Test-time scaling for tool-augmented agents expands both thinking (tokens) and acting (tool calls). While textual reasoning tasks are usually scaled by thinking with more tokens, solely increasing the internal thinking effort in agents proves to be suboptimal and does not always translate to performance scaling (Shen et al., 2025). Unlike controlling the token count in textual reasoning (Han et al., 2025b; Pu et al., 2025), in agent tasks such as web browsing, the number of tool calls directly determines the depth and breadth of exploration (Anthropic, 2025c; Team et al., 2025b), defining the effective boundary of external information access.

However, simply granting agents more test-time tool-call resources does not guarantee better performance. Our analysis reveals a critical bottleneck: *standard agents lack inherent budget awareness*. Without explicit signals, they often perform shallow searches (Lu et al., 2025) and fail to utilize additional resources, even when available. As our empirical findings later demonstrate, standard agents quickly hit a "performance ceiling", saturating their capabilities regardless of how much extra budget is allocated. The challenge is not simply spending more, but spending wisely: the marginal benefit per tool call is uncertain, so every call must be spent strategically.

Figure 1: Budget Tracker is a lightweight plug-in that can be applied to both a standard ReAct agent (top) and more advanced orchestration frameworks like BATS (bottom). In this figure, blue boxes highlight modules that adapt to the budget.

This unique complexity brings up a critical research question: How can tool-augmented agents scale *effectively* by making the best use of a given resource budget? We study this question in a budget-constrained setting, grounding our analysis in search agents equipped with search and browse tools, which are widely used in practice (Google, 2025; OpenAI, 2025) and inherently require extensive tool calls to collect external information. To ensure a fair and transparent comparison, we formalize a unified cost metric that jointly accounts for the economic costs of both internal token consumption and external tool interactions. This metric enables us to trace a true cost-performance scaling trend.

We first develop an intuitive, light-weight approach: **Budget Tracker** (Figure 1 (top)), which is a plug-and-play module compatible with most ReAct-based agents (Yao et al., 2023). Budget Tracker provides the agent with a continuous signal of resource availability, proving to be a simple yet effective method for enabling budget-aware tool use. Through extensive experiments, we show that this simple plug-in improves performance across various budget constraints. Combining this explicit budget awareness with traditional test-time scaling strategies consistently enables more effective scaling behaviors while continuously pushing the cost-performance Pareto frontier.

While simple awareness is effective, it still operates within the agent's predefined logic. To further break the scaling ceiling and fully internalize resource constraints, we introduce **BATS** (**B**udget **A**ware **T**est-time **S**caling) (Figure 1 (bottom)), a framework designed to maximize agent performance under any given budget. At its core, BATS maintains a continuous signal of remaining resources, and it uses this information to dynamically adapt its behavior. A planning module adjusts stepwise effort to match the current budget, while a verification module decides whether to "dig deeper" into a promising lead or "pivot" to alternative paths based on resource availability.

Our empirical study systematically evaluates the relationship between overall resource cost and task performance by comparing different scaling frameworks under varying budgets. The results show that BATS produces more favorable scaling curves: it achieves higher performance while using fewer tool calls and incurring lower overall cost than competing methods. These findings demonstrate the potential of budget-aware design for creating effective and efficient tool-augmented agents, highlighting the importance of explicitly accounting for cost in agent test-time scaling.

We summarize our contributions as follows:

- We provide the first systematic study of budget-constrained tool-use agents by formalizing agent test-time scaling with explicit tool-call budgets and introducing a unified cost metric.

- We introduce **Budget Tracker**, a light-weight, plug-in module compatible with any agent orchestration framework that enables effective budget-aware tool use.

- We develop **BATS**, a budget-aware framework that dynamically adapts planning and verification strategies based on real-time resource tracking, flexibly switching between deepening a lead and branching to alternatives.

- We conduct systematic experiments under varying budgets and unified costs with search agents, demonstrating that BATS is more cost-effective and yields more favorable scaling curves and better cost–performance trade-offs.

## 2 PROBLEM FORMULATION

### 2.1 AGENT TEST-TIME SCALING

We formulate agent test-time scaling as how an agent's performance scales with its budget for external tool-call interaction, refining the broader concept of test-time interaction scaling as discussed in Shen et al. (2025). To make agent test-time scaling cost-effective, an ideal agent should be able to achieve its best possible performance under an arbitrary budget constraint on the scaling curve. To this end, our target is to design an effective and efficient agent framework, $\pi$, that maximizes answer accuracy while adhering to a strict tool-call budget.

Assume the agent is equipped with a set of $K$ tools as $\mathcal{T} = \{t_1, \ldots, t_K\}$. For a given question $x \in \mathcal{X}$, the agent works under a budget $\mathbf{b} = (b_1, \ldots, b_K)$, where $b_i$ is the maximum number of invocations of tool $t_i \in \mathcal{T}$. Let $\hat{y}_\pi(x)$ denote the agent's predicted answer for question $x$ with ground truth $y(x)$ and let $c_i(x; \pi)$ be the realized number of calls to tool $t_i$ on $x$. We formulate the agent test-time scaling problem as a budget-constrained optimization objective:

$$
\begin{aligned}
\max_\pi \quad & \mathrm{Acc}_\mathbf{b}(\pi) \;\; = \;\; \mathbb{E}_x\big[\, \mathbf{1}\{\hat{y}_\pi(x) = y(x)\}\,\big] \\
\text{s.t.} \quad & c_i(x; \pi) \;\leq\; b_i \quad \text{for all } i = 1, \ldots, K, \text{ and every } x \in \mathcal{X}
\end{aligned}
\tag{1}
$$

Here the objective is the expected accuracy over all questions. The constraint ensures that for any given question, the number of realized calls for each tool never exceeds its allocated budget. By evaluating the agent performance at various budget levels, we can trace the performance-cost curve, which characterizes the agent's test-time scaling behavior, showing how effectively it leverages budget resources to its problem solving capabilities.

**Budget vs. Cost.** We distinguish between the preset budget constraint, which specifies the maximum number of tool calls available to the agent, and the realized cost, which reflects the resources actually consumed during execution. While the budget imposes a hard upper limit, the final cost depends on the agent's strategy. To facilitate a more consistent and comprehensive comparison, we introduce in Section 2.2 a unified post-hoc cost metric for analyzing agents' test-time scaling.

**Choice of Budget.** Among possible constraints, we prioritize a tool-call budget over a token-based budget for its *relevance*, *consistency*, and *practicability*. A tool-call limit offers a more relevant and direct constraint on an agent's ability to acquire external knowledge than the tokens used for internal reasoning. This choice is also consistent with and justified by established practices (Shen et al., 2025; Team et al., 2025b). Furthermore, it offers greater practicability, as it is often non-trivial to pre-determine an appropriate token budget for complex, multi-step agentic tasks. While the budget is defined by tool calls, we still incorporate token usage into our unified cost metric (Section 2.2) to ensure a more fair and transparent comparison.

### 2.2 PROBLEM INSTANTIATION WITH SEARCH AGENT

In our work, we instantiate the test-time scaling problem with search agent, a setting selected for its broad applicability and the presence of established benchmarks.

A search agent is an LLM that answers an information-seeking question $x$ by retrieving external evidence and reasoning over it. The agent follows an iterative ReAct-style loop (Yao et al., 2023), alternating between internal thinking and external actions. The agent has access to two primary tools for interacting with the world:
**(1) Search.** This tool helps perform a standard search engine query. Given a text query, it returns a list of search results, each including a title, a brief snippet, and a URL.
**(2) Browse.** Given a specific URL, this tool scrapes the full content of the corresponding webpage, providing detailed information that is often unavailable in a search snippet.

**Unified Cost Metric.** We model the agent's total cost as the sum of consumed resources along two dimensions: tokens and tool calls. To create a unified metric, we map both to their corresponding economic costs.
**(1) Token cost.** This represents the agent's internal cognitive effort, including its reasoning, planning, and parametric knowledge processing. Token costs are calculated based on the pricing of the model provider, distinguishing among input, output, and cache hit tokens. In multi-round iterative

frameworks, the output of iteration $i$ becomes part of the input for iteration $i + 1$. Any overlap is a cache hit, thereby lowering the token consumption cost.

**(2) Tool call cost.** This represents the agent's active interaction with the external environment through information-seeking actions. Each invocation of an external service (e.g., a search query or a browsing request) incurs a cost determined by the pricing of the corresponding API provider.

The total unified cost, $C_{unified}(x; \pi)$, for solving a given question $x$ under policy $\pi$ is the sum of the token cost and the tool call cost. Let $c_i(x; \pi)$ be the number of actual invocations to tool $t_i$ and $P_i$ be the economic cost per invocation of $t_i$. The unified cost metric is then defined as:

$$C_{unified}(x; \pi) = \underbrace{c_{token}(x; \pi)}_{\text{Token Cost}} + \underbrace{\sum_{i=1}^{K} c_i(x; \pi) \cdot P_i}_{\text{Total Tool Cost}} \tag{2}$$

Here, $c_{token}(x; \pi)$ represents the total cost incurred from token consumption during the agent's reasoning process for question $x$. This formulation allows us to uniformly analyze the actual incurred costs for each execution. These two dimensions are inherently coupled: additional tool calls generally increase token consumption, as the agent must process and reason over the retrieved external information. Measuring this unified cost under varying budget constraints, $\mathbf{b}$, enables a more comprehensive and consistent analysis across different policies.

## 3 BUDGET AWARENESS

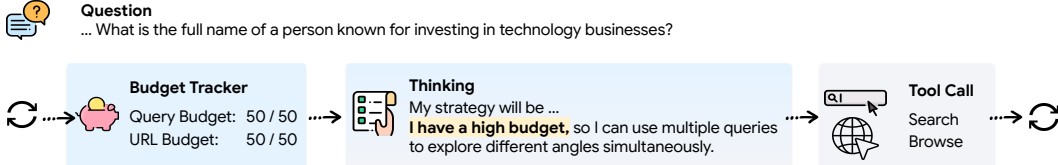

Figure 2: At each interaction round, the agent is provided with its current and remaining budget through the budget tracker before generating the next thinking step and the tool call actions.

### 3.1 BUDGET TRACKER

Each tool invocation introduces non-negligible costs in context and computation. We hypothesize that providing explicit budget signals enables the model to internalize resource constraints and adapt its strategy without requiring additional training. We introduce **Budget Tracker**, a lightweight, plug-and-play module that surfaces real-time budget states inside the agent's reasoning loop. In the beginning iteration, the tracker provides a brief policy guideline describing the budget regimes and corresponding tool-use recommendations (see Appendix F). At each subsequent iteration, this tracker appends a budget status block showing the remaining and used budgets for each available tool(see Appendix A). Despite its simplicity, Budget Tracker operates purely at the prompt level and makes the agent explicitly aware of its resource consumption and remaining budget, enabling it to condition subsequent reasoning steps on the updated resource state.

### 3.2 SCALING WITH BUDGET AWARENESS

To better understand how agent behaviors scale under different budget constraints, we study two mainstream test-time scaling paradigms: *sequential scaling* and *parallel scaling*. In *sequential scaling*, we adopt the budget-forcing strategy (Muennighoff et al., 2025) by appending the following message when the agent proposes an answer: "Wait, you still have remaining tool budget, use more search and browse tools to explore different information sources before concluding." This encourages the model to reevaluate its reasoning and make better use of its available tool calls. In *parallel scaling*, we fix the per-run budget and conduct multiple independent runs in parallel. We report Majority Vote, Best-of-N and Pass@N results, following common test-time scaling practice (Wei et al., 2025). More details can be found in Appendix E.2.

## 3.3 RESULTS AND ANALYSIS

We evaluate Budget Tracker on three information-seeking QA datasets requiring external search: BrowseComp (Wei et al., 2025), BrowseComp-ZH (Zhou et al., 2025) and HLE-Search (Phan et al., 2025; Han et al., 2025a). Dataset details and full experimental setup are provided in Section B. Building on the ReAct framework, we incorporate the Budget Tracker immediately after each tool response to inform the agent of its remaining budget. Each tool is assigned a budget of 100, and the reasoning process stops when any budget is exhausted or the agent reaches a final answer.

Table 1: Effect of Budget Tracker on tool-use behavior and performance. Adding the tracker consistently improves answer accuracy across models and datasets. Results are averaged across 3 runs.

| Model | Method | BrowseComp | BrowseComp-ZH | HLE-Search |
|---|---|---|---|---|
| Gemini-2.5-Pro | ReAct | 12.6 | 31.5 | 20.5 |
| | + Budget Tracker | **14.6** | **32.9** | **21.8** |
| Gemini-2.5-Flash | ReAct | 9.7 | 26.5 | 14.7 |
| | + Budget Tracker | **10.7** | **28.7** | **17.3** |
| Claude-Sonnet-4 | ReAct | 12.2 | 29.1 | 20.5 |
| | + Budget Tracker | **14.0** | **31.1** | **23.0** |

**Budget Tracker enables higher performance under the same budget.** Table 1 compares ReAct with and without the Budget Tracker under identical budget limits. Across different models, adding the tracker consistently improves accuracy on all the datasets. This demonstrates that making budget signals explicit encourages more strategic and positive agent tool-use behavior.

**Budget Tracker achieves lower resource usage and cost under similar accuracy.** In Table 2, the consumed resources report the average number of search and browse tool calls, as well as the unified cost combining tool and token consumption per question. Increasing the tool-use budget in ReAct improves performance but also raises both tool call frequency and total cost. In contrast, Budget Tracker achieves comparable accuracy with less budget (10 vs. 100), while using 40.4% fewer search calls, 21.4% fewer browse calls, and reducing overall cost by 31.3%. This demonstrates that explicit budget awareness enables more efficient tool use without sacrificing accuracy.

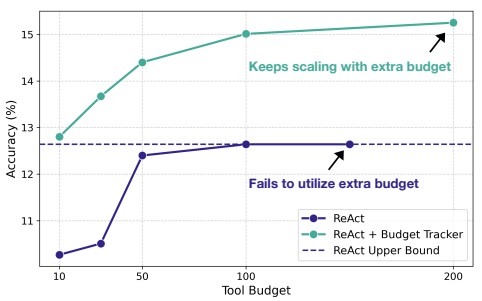

Figure 3: ReAct saturates and fails to utilize additional tool budget, reaching a performance ceiling. In contrast, ReAct + Budget Tracker continues to scale effectively with larger budgets, achieving consistent accuracy improvements.

**Budget Tracker facilitates continued scaling through effective resource utilization by simply adding budget awareness.** Figure 3 illustrates Gemini-2.5-Pro's performance on BrowseComp as tool budget increases. Lacking budget awareness, the standard ReAct baseline saturates at a budget of 100. Beyond this point, it fails to utilize any additionally allocated budget, even though the context window is not filled up. This ceiling occurs because the agent's reasoning process terminates prematurely: it either believes it has found a sufficient answer or concludes it is stuck and gives up, unaware of the unused resources. In contrast, Budget Tracker explicitly informs the model of its remaining resources, enabling it to leverage larger budgets and continue scaling its performance.

## 3.4 ANALYSIS ON TEST-TIME SCALING STRATEGIES

Across different test-time scaling strategies, **Budget Tracker consistently pushes the Pareto frontier and demonstrates robust effectiveness.** Sequential scaling encourages the agent to issue more tool calls by prompting it to continue reasoning and exploring additional information sources. The prompting stops once the agent refuses to invoke any tools for five iterations. As shown in Figure 4,

Table 2: Performance and resource consumption comparison under different budgets using Gemini-2.5-Pro. With 10× less budget (10 vs. 100), Budget Tracker achieves comparable accuracy while requiring fewer tool calls and lower total cost, highlighting the benefits of explicit budget awareness.

| Methods | Budget | Acc (%) | Consumed resources | | |
|---|---|---|---|---|---|
| | | | # search | # browse | Unified cost (¢) |
| ReAct | 10 | 10.3 | 7.87 | 0.60 | 5.2 |
| | 30 | 10.5 | 13.77 | 1.33 | 8.0 |
| | 100 | 12.6 | 14.24 | 1.36 | 9.9 |
| **ReAct + Budget Tracker** | 10 | **12.8** | 8.48 | 1.09 | 6.8 |

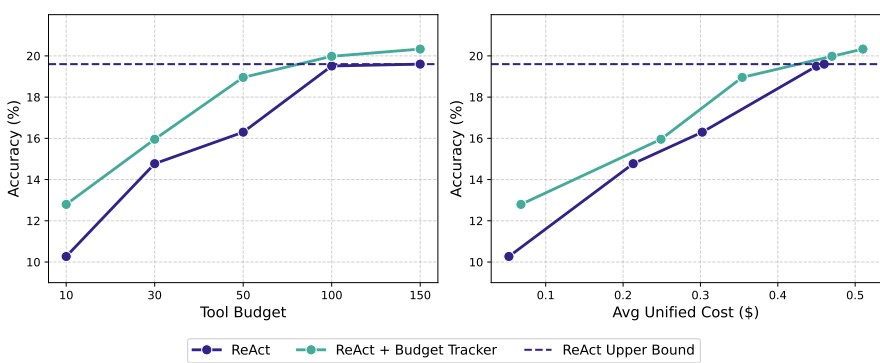

Figure 4: Comparison of Budget Tracker and ReAct in sequential scaling using Gemini-2.5-Pro. With explicit budget awareness, Budget Tracker consistently improves upon ReAct at equal budgets. ReAct plateaus early as it cannot utilize extra resources, while Budget Tracker adapts its spending to gain further improvements and extend the cost-performance frontier.

while this strategy enables ReAct to further improve accuracy, it eventually encounters a performance ceiling, where the agent confidently concludes that no further tool calls are necessary. In contrast, Budget Tracker achieves a consistently better scaling curve and sustains performance gains beyond ReAct's plateau. Furthermore, it advances the Pareto frontier of the cost–performance curve, demonstrating not only more effective performance scaling but also more efficient tool utilization.

For parallel scaling, we set a fixed budget of 50 per run for Gemini-2.5-Pro, as empirical observations indicate that only 0.8% of the data in BrowseComp requires exceeding this limit (see details in Appendix E.1). In Figure 5, we use identical sample sizes for both settings and report Majority Vote and Best-of-N results (Pass@N results are provided in Appendix E.2). On the right, the figure illustrates the cost–performance scaling, where Budget Tracker consistently yields a superior curve.

## 4 BATS: BUDGET-AWARE AGENT TEST-TIME SCALING

Given the demonstrated effectiveness of budget awareness, we now explore how it enhances agent orchestration and influences key behaviors such as planning and self-verification. In this section, we propose **BATS**, a **B**udget-**A**ware **T**est-time **S**caling framework for tool-augmented agents under explicit budget. As shown in Figure 6, given an information-seeking question and a tool call budget, BATS begins with internal reasoning to formulate a structured action plan and decide which tools to invoke. Tool calls are triggered through specific tokens, and the returned tool responses are appended to the reasoning sequence, expanding the context with new evidence. When the agent proposes a candidate answer, the verification module checks its validity and decides whether to continue the current sequence or initiate a new attempt with the remaining budget. The iterative process terminates once any budgeted resource is exhausted. Finally, an LLM-as-a-judge selects the best answer across all verified answers. The full prompts of BATS are provided in Appendix F.

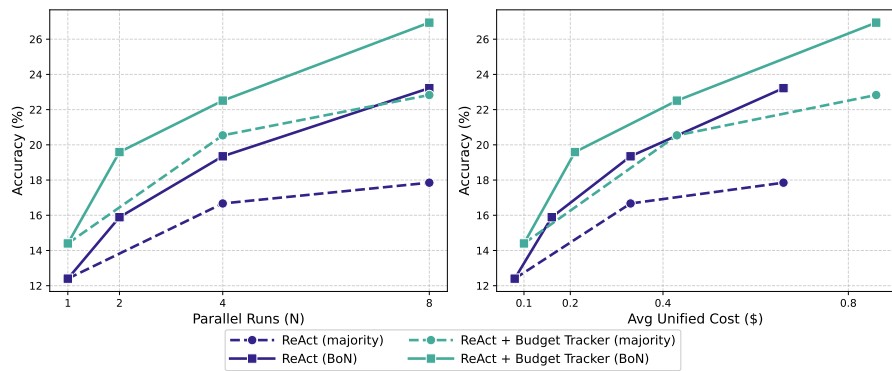

Figure 5: Comparison of Budget Tracker and ReAct in parallel scaling using Gemini-2.5-Pro. The left subfigure shows accuracy scaling with increasing parallel runs, while the right subfigure illustrates the corresponding cost–performance trend.

The central design principle of BATS is *budget awareness*. Throughout the execution, the Budget Tracker continuously updates both resource usage and remaining budget at every iteration. This persistent awareness shapes the agent's planning, tool-use strategy, and verification behavior, enabling adaptive and efficient use of constrained resources. We next describe how each module in BATS incorporates budget awareness.

### 4.1 BUDGET-AWARE PLANNING

Planning in BATS incorporates both *constraint decomposition* and *structured dynamic planning*. Selecting an appropriate starting point is critical: a well-chosen entry narrows the search space and conserves budget, while a poor choice can quickly exhaust the budget (see example in Appendix G.1). To address this, we prompt the agent to first perform **constraint decomposition** and to categorize the clues implied in the question into two types: (1) *exploration*, which expand the candidate space, and (2) *verification*, which validate specific properties. While a verification clue can sometimes provide a direct shortcut to the answer, relying on it prematurely is risky, as it may consume resources without guaranteeing progress.

The agent is further instructed to generate and maintain an explicit plan throughout execution. This **tree-structured plan** acts as a dynamic checklist, recording step status, resource usage, and allocation, while guiding future actions. It is never overwritten: completed, failed, or partial steps remain recorded to prevent redundant tool calls and to maintain a full execution trace. We provide the full prompt of planning module in Appendix F.2. As shown in the planning block from Figure 6, a single step in the plan represents a subtask that requires multiple tool calls to complete, for instance, getting a candidate list. After every iteration, new information may create additional branches, resolve pending steps, or invalidate unproductive paths. The planning module adjusts exploration breadth and verification depth based on the current remaining budget.

This integration of constraint decomposition, persistent structured planning, and continuous budget-conditioned updating allows BATS to maintain a controlled and interpretable search process while efficiently allocating available tool calls across exploration and verification subtasks.

### 4.2 BUDGET-AWARE SELF-VERIFICATION

Once the agent proposes an answer, the self-verification module re-evaluates the reasoning trajectory and corresponding resource usage. The full verifier prompt is provided in Appendix F.3. This process begins with a constraint-by-constraint backward check. Using the previously extracted exploration and verification clues, the module assesses each constraint and determines whether it has been satisfied, contradicted, or remains unverifiable. This detailed check grounds the proposed answer directly against the question's requirements and clues.

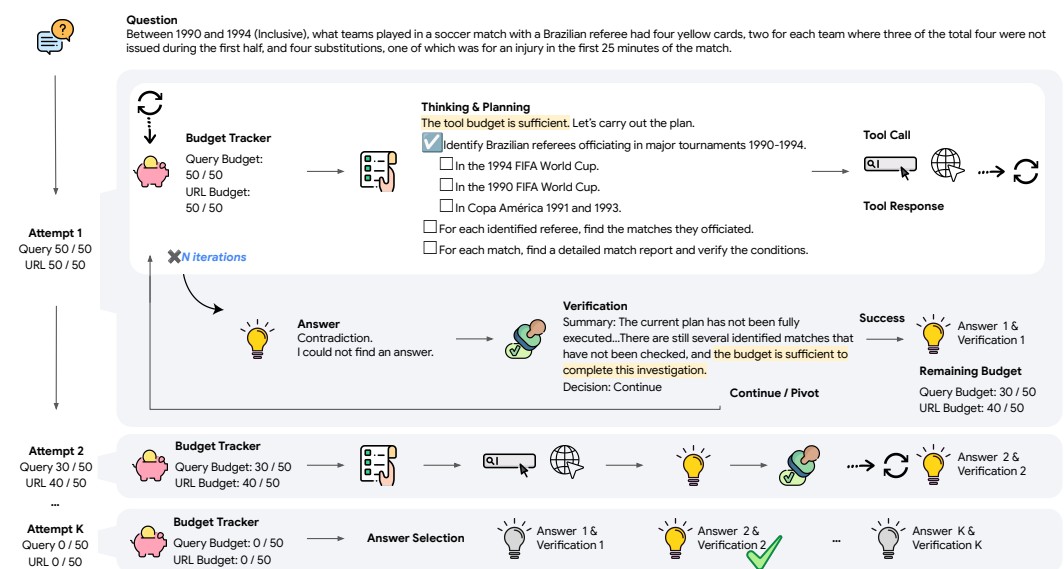

Figure 6: Overview of the BATS framework. Given a question and per-tool budgets, the agent begins with budget-aware thinking and planning, structured as a checklist. The agent keeps iterating by reasoning over new information and updated budgets. When an answer is proposed, BATS performs verification and decides to either continue, pivot, or initiate a new attempt with the remaining budget. BATS terminates when any of the budgets are exhausted.

Based on this analysis, the module then makes a verification decision based on the above assessment and budget status. If all constraints are satisfied, the answer is marked as a *SUCCESS*. If several constraints remain unverifiable but the trajectory appears promising, provided the budget is sufficient for deeper exploration, the outcome is to *CONTINUE* exploration. In contrast, if contradictions are identified or the remaining budget cannot support further investigation towards this lead, the agent is expected to terminate expensive or low-yield directions early and *PIVOT* toward a different direction to avoid wasting tool call resources while resources are still sufficient to pursue alternatives. When the decision is to *CONTINUE* or *PIVOT*, the module also generates a concise summary that replaces the raw trajectory in context. This includes key reasoning steps, intermediate findings, failure causes, and suggestions for optimization to avoid redundant exploration. By compressing the reasoning trajectory into a compact and informative summary, the verifier reduces context length while ensuring that subsequent attempts remain grounded in previously acquired information.

Together, structured constraint checking, explicit decision rules, and budget-aware trajectory summarization allow BATS to terminate useless trajectories early, continue promising ones efficiently, and maintain reliable progress toward the correct answer within strict budget constraints.

## 5 EXPERIMENTS AND RESULTS

We use three challenging information-seeking benchmarks: BrowseComp (Wei et al., 2025), BrowseComp-ZH (Zhou et al., 2025) and HLE-Search (Han et al., 2025a). More experiment details can be found in Appendix B.

### 5.1 MAIN RESULTS

Table 3 shows the performance comparison across different web search agents. Under the strict budget constraints of 100 tool uses for BrowseComp, BATS consistently achieves better results than baselines, obtaining 24.6% on BrowseComp, 46.0% on BrowseComp-ZH and 27.0% on HLE-Search using Gemini-2.5-Pro. Crucially, while many comparing agents rely on extensive task-specific training to boost performance, BATS is entirely training-free. This result highlights the ef-

Table 3: Performance comparison across web search agents. We denote results from our own experiments with ⋆; other baseline scores are cited from their respective publications. The "Training" column specifies whether the model has been specifically trained on agentic web search tasks. For our budget-constrained setting, each agent is provided a budget of 100 tool uses per tool.

| Method | | Training | BrowseComp | BrowseComp-ZH | HLE-Search |
|---|---|---|---|---|---|
| *Model Only* | | | | | |
| GPT-4o | | ✗ | 0.6 | 6.2 | - |
| Claude-3.7-Sonnet | | ✗ | 2.3 | 11.8 | - |
| Gemini-2.5-Flash⋆ | | ✗ | 2.7 | 23.9 | 2.8 |
| Gemini-2.5-Pro⋆ | | ✗ | 6.3 | 27.8 | 8.6 |
| OpenAI o1 | | ✗ | 9.9 | 29.1 | - |
| *Training-based Agents* | | | | | |
| ASearcher | | ✓ | 5.2 | 15.6 | - |
| WebSailor | | ✓ | 12.0 | 30.1 | - |
| DeepDive | | ✓ | 14.8 | 25.6 | - |
| WebExplorer | | ✓ | 15.7 | 32.0 | - |
| OpenAI Deep Research | | ✓ | **51.5** | **42.9** | **29.1** |
| *Budget-constrained* | | | | | |
| Claude-Sonnet-4 | ReAct | ✗ | 12.2 | 29.1 | 20.5 |
| Claude-Sonnet-4 | BATS (Ours) | ✗ | 19.1 | 41.5 | **29.0** |
| Gemini-2.5-Flash | ReAct | ✗ | 9.7 | 26.5 | 14.7 |
| Gemini-2.5-Flash | BATS (Ours) | ✗ | 14.3 | 34.3 | 19.5 |
| Gemini-2.5-Pro | ReAct | ✗ | 12.6 | 31.5 | 20.5 |
| Gemini-2.5-Pro | BATS (Ours) | ✗ | **24.6** | **46.0** | 27.0 |

fectiveness of our budget-aware design, which maximizes the efficiency of every tool call to achieve superior results in resource-constrained settings without the need for additional fine-tuning.

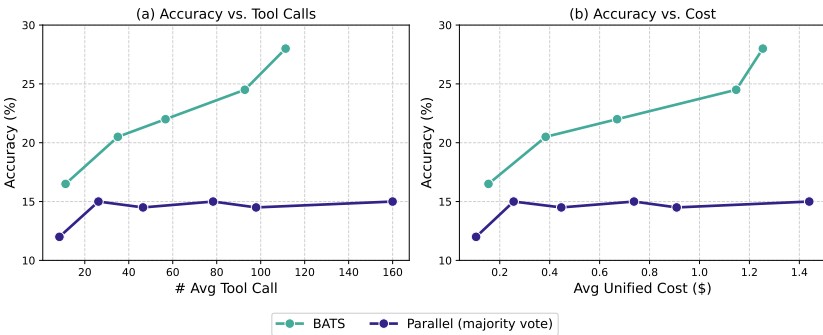

Figure 7: Scaling behaviors along (a) total number of tool calls and (b) average unified cost, evaluated on a 200-example subset of BrowseComp using Gemini-2.5-Pro.

**BATS achieves higher performance under the same budget constraint.** To better understand the scaling behavior, we vary the tool-call budget and evaluate performance on a random subset of 200 examples from BrowseComp for a manageable analysis. Figure 7 (left) shows the accuracy against the average number of tool calls, including both search and browse. Across all budget levels, BATS consistently outperforms the parallel majority-vote baseline, demonstrating that budget-aware adaptation leads to more effective use of limited tool calls.

**BATS achieves higher performance when accounting for unified costs.** Beyond tool-call counts, we measure performance under a unified cost metric that incorporates both token usage and tool-call expenses. As shown in Figure 7 (right), BATS achieves more favorable scaling curves, delivering higher accuracy at comparable or lower costs. This indicates that BATS not only improves effectiveness under budget constraints but also yields better cost–performance trade-offs.

## 5.2 EARLY STOPPING

We analyze how effectively agents can solve questions under budget constraints when allowed to stop early, without exhausting all tool calls. This setting reflects realistic scenarios where efficient, confident answers are preferable to prolonged exploration. To isolate this behavior, we focus on the agent's *first attempt* only. For BATS, this is the first answer that successfully passes self-verification. For baselines that lack verification, we use the output from a single generation pass as their first attempt. If any tool budget is exhausted before this first attempt completes, the agent will stop and output a final answer based on its progress so far. This evaluation captures both efficiency and robustness, testing whether agents can solve questions quickly without using their full budgets.

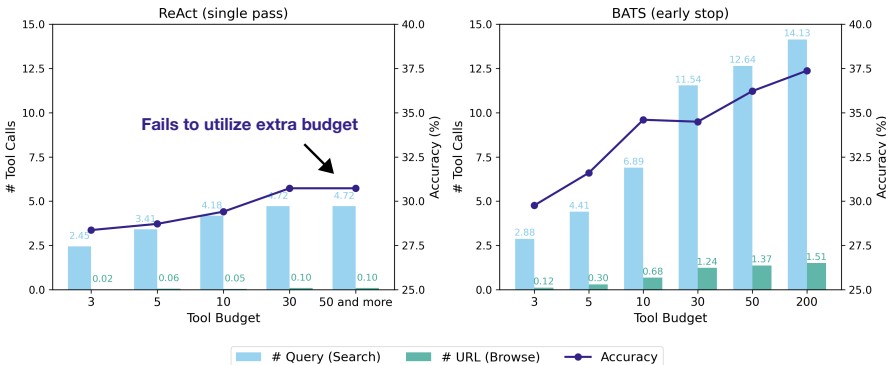

Figure 8: Performance and resource utilization under early stopping on BrowseComp-ZH using Gemini-2.5-Pro. The bars represent the average number of Search and Browse tool calls (left axis), while the line plots indicate accuracy (right axis). While ReAct plateaus and fails to utilize additional budget, BATS effectively scales up tool usage to achieve higher accuracy as the budget increases.

**Budget awareness enables BATS to scale effectively with increased resources.** Figure 8 presents the early stopping performance on BrowseComp-ZH using Gemini-2.5-Pro. The x-axis denotes the predefined tool call budget for both search and browse tool calls. Bars show the average number of tool calls actually used, while the line plot reports the resulting accuracy. The ReAct baseline (Figure 8 left) exhibits poor budget utilization: it consistently underuses the browse tool (fewer than 0.1 calls on average) and reaches a performance plateau early, with accuracy capped at 30.7% for all budgets of 30 and above. This behavior highlights its lack of budget awareness and its inability to benefit from additional resources. In contrast, BATS (Figure 8 right) shows strong budget-adaptive behavior. As the budget increases, it strategically raises its use of both search and browse tools, leading to steady gains in accuracy, rising from 29.8% (budget=3) to 37.4% (budget=200). Notably, with a budget of just 5, BATS already surpasses the baseline's best achievable accuracy. This demonstrates BATS's ability to make more strategic and cost-effective decisions, delivering higher accuracy even with the same or fewer resources. We provide ablation studies in Appendix C. More analysis can be found in Appendix E.4.

## 6 CONCLUSION

In this work, we present the first systematic study of budget-constrained tool-use agents and their test-time scaling behaviors. We identify that standard agents often hit a performance ceiling due to a lack of resource awareness. To overcome this, we introduce the **Budget Tracker**, a lightweight plug-in that provides budget awareness, and **BATS**, a comprehensive framework that dynamically adapts planning and verification based on real-time resource status. Extensive experiments across multiple models and information-seeking benchmarks show that budget awareness enables stronger agent scaling and consistently pushes the cost-performance Pareto frontier. By formalizing tool-call budgets as a critical scaling dimension, our work offers a principled foundation for building efficient and adaptive agents.

## THE USE OF LARGE LANGUAGE MODELS (LLMS)

We acknowledge the use of LLMs (ChatGPT and Gemini) exclusively for editing the text to correct grammatical errors and improve clarity and flow. All core scientific content and research ideas were authored solely by the human authors.

## REPRODUCIBILITY STATEMENT

We conduct evaluations exclusively on publicly available benchmarks and query LLMs through public providers. All experimental settings, including temperature, context length, tool calling, configuration parameters, and prompts, are detailed in Appendix B and F.

## ETHICS STATEMENT

The authors confirm adherence to the ICLR Code of Ethics. This work aims to reduce the computational and economic costs of LLM agents, contributing positively to accessibility and sustainability. At the same time, we recognize that web search agents may inherit web-based biases or propagate misinformation. Addressing these risks requires our continued attention and responsible research practices within the research community.

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

## A  BUDGET TRACKER

> Budget Tracker
>
> <budget>
> Tool$_1$ Budget Used: ##, Tool$_1$ Budget Remaining: ##
> Tool$_2$ Budget Used: ##, Tool$_2$ Budget Remaining: ##
> Make the best use of the available resources.
> < /budget>

## B  EXPERIMENT DETAILS

### B.1  SETUP

**Datasets.** To evaluate web search agents, we use three challenging information-seeking benchmarks: BrowseComp (Wei et al., 2025), a dataset of 1,266 difficult web-browsing questions requiring persistent retrieval; BrowseComp-ZH (Zhou et al., 2025), a 289-question Chinese benchmark designed to test agents' performance in region-specific web environments; and HLE-Search (Han et al., 2025a), a curated 200-question subset of Human's Last Exam (Phan et al., 2025) focusing on queries that explicitly require search rather than pure reasoning.

**Baselines.** We compare BATS against a range of models and agentic frameworks, including both general-purpose base models (Hurst et al., 2024; Jaech et al., 2024; Comanici et al., 2025; Anthropic, 2025a;b) and those specifically fine-tuned for agentic search tasks (Li et al., 2025b; Liu et al., 2025a; Gao et al., 2025; Lu et al., 2025). To evaluate the final answer accuracy, we use Gemini-2.5-Flash as the judge model and adopt the evaluation prompt from Phan et al. (2025). Baseline results for HLE-Search are scraped from Han et al. (2025a).

For scaling methods, we evaluate sequential and parallel scaling approaches applied to the ReAct (Yao et al., 2023) baseline. For **sequential scaling**, to encourage the agent to use more tools during its iterative execution, we follow the approach from textual reasoning (Muennighoff et al., 2025) and append the instruction that encourages it to rethink and use more tools whenever the agent provides an answer. This process is repeated until the agent's tool budget is exhausted, after which it is prompted to produce the final answer. For **parallel scaling**, we use temperature sampling to generate diverse reasoning paths. To aggregate the results, by default, we use Gemini-2.5-Flash

to select the most common answer via a majority vote (Wang et al., 2023) (see Appendix E.2 for details). For experiments in BATS, to enforce the budget constraint for each question, we continue sampling new sequences until the budget is fully consumed. Thus the number of sampled sequences may differ across questions.

## B.2 IMPLEMENTATION DETAILS

We use Gemini-2.5-Flash, Gemini-2.5-Pro (Comanici et al., 2025) and Claude-Sonnet-4 (Anthropic, 2025b) as the backbone models in our framework. By default, we disable the thinking mode by setting the thinking budget as 0 for Gemini-2.5-Flash and 1024 for Gemini-2.5-Pro models. The maximum number of new tokens for generation was set to 65,536 for Gemini models and 64,000 for Claude. We use a temperature of 0.7 during agent execution to encourage exploration, and use a deterministic temperature of 0.0 for final answer selection and answer evaluation. We use the Google Custom Search JSON API[1] for search tools, Jina.ai[2] and Crawl4AI (UncleCode, 2024) for web browsing.

To keep the context length manageable, we employ several simple context management strategies. For each browse tool call, the fetched webpage is truncated to 150,000 characters before being sent to the model. At every iteration, we discard tool outputs from previous steps and retain only the most recent tool response, preventing the context from growing with accumulated tool results. In BATS's verification module, we further control context size by periodically replacing the historical trajectory with summary. Since the agent determines when to activate the verification module, we perform a check during each invocation: if more than $K$ iterations have passed since the last update (with $K = 10$ in our experiments), we replace the older reasoning trace with a concise summary derived from the verification outputs.

## B.3 RESOURCE COST

The cost of tool calls is determined by the pricing of the providers. To standardize billing, we've established a unified rate of \$0.001 per invocation for both search API calls and web browsing actions. Because different URLs produce varying amounts of text and tokens, this per-call rate is derived as an average computed from post hoc statistics over all our experiments. The consumption of tokens is billed separately, adhering to the official pricing models of the API provider pricing[3].

## C ABLATIONS

Table 4: Ablation results of BATS on three benchmarks with Gemini-2.5-Pro. The agent is provided a budget of 100 tool uses per tool. Removing different modules leads to various performance drops across datasets.

| Method | BrowseComp | BrowseComp-ZH | HLE-Search |
|---|---|---|---|
| BATS | **18.7** | **39.1** | **23.0** |
| w/o Planning | 17.0 | 34.6 | 20.0 |
| w/o Verification | 15.4 | 37.7 | 22.0 |
| w/o Planning & Verification | 14.6 | 32.9 | 21.5 |

In this section, we present an ablation analysis of the key modules within BATS using Gemini-2.5-Pro. The full comparison is shown in Table 4. Across all settings and datasets, the agent is allocated a budget of 100 tool uses per tool. To provide a clearer view of the orchestration framework, we focus on BATS under the early-stopping mechanism, where generation terminates once an answer is verified as successful.

Results indicate that removing the planning module leads to a moderate performance decrease. However, removing the verification module causes a more significant drop on BrowseComp (from 18.7%

---

[1] https://developers.google.com/custom-search/v1/overview
[2] https://jina.ai/
[3] https://cloud.google.com/vertex-ai/generative-ai/pricing

to 15.4%). This suggests that the verification module is crucial for enabling the agent to navigate the solution space and accurately assess its current progress. Removing both modules results in lower performance (14.6% on BrowseComp), while BrowseComp-ZH and HLE-Search exhibit some variance, likely due to the smaller dataset size. In addition, questions in HLE-Search are typically shorter and contain fewer details that can be verified, which limits the contribution of the verification module. Overall, the trend confirms that both modules contribute positively to the orchestration framework's effectiveness.

# D    RELATED WORK

## D.1    TEST-TIME SCALING

Test-time scaling (TTS) (Snell et al., 2024; Zhang et al., 2025a) strategies typically fall into two categories. The first is sequential scaling, where a model iteratively refines its output based on self feedback or reflection (Madaan et al., 2023; Zhang et al., 2025b; Muennighoff et al., 2025; Liu et al., 2025b). The second category is parallel scaling, where multiple reasoning paths are sampled and an aggregation strategy is used to determine the final answer (Brown et al., 2024; Wang et al., 2023). Further, hybrid scaling attempts to combine their complementary benefits (Chen et al., 2025a;b; Wan et al., 2025; Li et al., 2024). While prior work focuses on text-only reasoning, we extend TTS to tool-augmented agents, where scaling accounts for both tokens and tool calls under budget constraints. As these methods push performance by increasing computation, a complementary line of work examines how to constrain the effort. Typical constraints are defined over tokens, sampled sequences, or FLOPs (Nayab et al., 2024; Welleck et al., 2024; Damani et al., 2025; Pu et al., 2025). Specifically, AgentTTS (Wang et al., 2025a) optimizes LLM size and sampling numbers under a unified FLOPs budget, while SLIM (Yen et al., 2025) uses periodic summarization to manage context growth in long-horizon agents. In contrast, we formalize and constrain the tool-call budget, shifting the focus from token-related limits in text reasoning to budget-constrained scaling of tool-augmented agents.

## D.2    WEB SEARCH AGENTS

Web search agents use search and browse tools to solve complex, multi-hop queries (Chen et al., 2025c; Wong et al., 2025; Team et al., 2025a; Han et al., 2025a; Li et al., 2025a). One research direction builds training data and applies various training methods to specialize the models (Jin et al., 2025; Wu et al., 2025a; Li et al., 2025b; Liu et al., 2025a; Tao et al., 2025; Ye et al., 2025; Li et al., 2025c). MiroThinker (Team et al., 2025b) shows scaling interactive tool use is an effective dimension, but it often yields redundant or inefficient tool calls. Another explores inference-time strategies (Li et al., 2025d; Zhu et al., 2025a; Qiao et al., 2025; Qin et al., 2025), such as incorporating programmatic execution to perform multiple tool call actions (Pang et al., 2025), finding an optimal, statically efficient configuration (Wang et al., 2025b), or exploring various design choices when scaling test-time compute (Zhu et al., 2025b). Instead, our work focuses on dynamic, cost-effective performance, providing the first analysis of agent scaling behavior under explicit budget constraints.

# E    ADDITIONAL ANALYSIS

## E.1    TOOL USAGE

In Table 5, we examine how the ReAct baseline utilizes tools under different budget constraints on the BrowseComp dataset. As expected, increasing the allowed tool budget enables the agent to interact more with the environment, leading to improved accuracy. The Over-Budget% column shows the proportion of examples that exhaust the allocated tool budget. For Gemini-2.5-Pro, only 0.8% of the data requires more than 50 tool calls, whereas Gemini-2.5-Flash shows a higher reliance on tool usage, with 2.6% of the queries exceeding a budget of 100. Notably, Flash models issue roughly twice as many search queries as Pro models. This suggests that the stronger parametric knowledge of Gemini-2.5-Pro allows it to navigate the search space more efficiently, reducing reliance on externally gathered evidence and enabling it to solve queries with fewer tool calls.

Table 5: Resource usage statistics of the ReAct baseline using Gemini-2.5-Pro and Gemini-2.5-Flash on the BrowseComp dataset. Over-Budget % denotes the percentage of data that reaches the given tool budget limit.

| Model | Budget | Acc % | Query | | URL | |
|---|---|---|---|---|---|---|
| | | | Avg. # | Over-Budget % | Avg. # | Over-Budget % |
| | 30 | 10.5 | 13.77 | 9.24 | 1.33 | 0.00 |
| Gemini-2.5-Pro | 50 | 12.4 | 13.95 | 0.79 | 1.31 | 0.00 |
| | 100 | 12.6 | 14.24 | 0.00 | 1.36 | 0.00 |
| | 30 | 9.0 | 22.77 | 42.81 | 1.04 | 0.00 |
| Gemini-2.5-Flash | 50 | 9.7 | 28.88 | 14.22 | 1.24 | 0.00 |
| | 100 | 10.0 | 29.93 | 2.6 | 1.36 | 0.00 |

## E.2 DETAILS OF PARALLEL SCALING

For Majority Vote, we aggregate the final answers across different samples and use a judge model (Gemini-2.5-Flash) to identify the consensus answer. For Best-of-N, we provide all response trajectories to the judge model and select the most promising one. For Pass@N, the score is calculated as 1 if any of the sampled responses contain the correct answer. The prompts used for these evaluations are provided in Appendix F.4. In Figure 9, we additionally report the Pass@N results for ReAct and Budget Tracker under parallel scaling settings. Budget Tracker consistently achieves higher overall accuracy across varying budgets and cost levels.

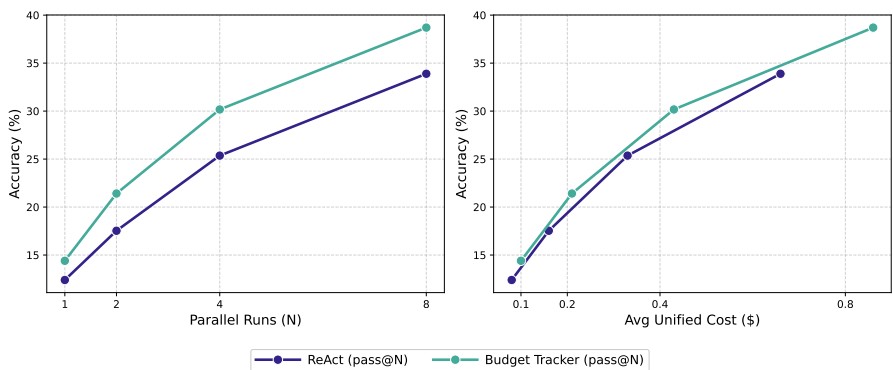

Figure 9: Pass@N results of Budget Tracker and ReAct in parallel scaling. The left subfigure shows accuracy scaling with increasing parallel runs, while the right subfigure illustrates the corresponding cost–performance trend.

## E.3 RESOURCE USAGE

Table 6 reports the detailed breakdown of tool calls and token consumption in BATS using Gemini-2.5-Pro on BrowseComp. We compare our approach against the variant BATS-response, which keeps the full tool responses from previous rounds. In contrast, BATS removes these responses to optimize context length. In the table, token usage is reported as a triplet: (Input / Output / Cache). The results demonstrate that removing historical tool responses does not compromise performance (maintaining similar accuracy) but significantly reduces computational overhead. This efficiency is evidenced by the substantial decrease in cache tokens and the reduction in overall cost.

## E.4 PLANNING MODULE

To evaluate the impact of the planning module, we augment the ReAct baseline with our proposed design, which integrates constraint analysis and a dynamically structured checklist plan. The tool-call budget is capped at 200 for both search queries and browse URLs. As shown in Table 7, the

Table 6: Resource usage comparison. Token counts are reported as Input / Output / Cache (in 10,000). BATS achieves comparable accuracy to the full-context baseline (BATS-response) while significantly reducing cache token usage and cost.

| | Acc (%) | # search | # browse | Total tokens | Veri. tokens | Cost ($) |
|---|---|---|---|---|---|---|
| BATS | 24.6 | 87.3 | 13.6 | 32.1 / 6.9 / 39.3 | 1.1 / 1.2 / 7.6 | 1.1 |
| BATS-response | 24.3 | 84.4 | 14.6 | 33.6 / 6.5 / 91.8 | 1.0 / 1.3 / 17.1 | 1.2 |

addition of planning module alone improves the agent's ability to organize exploration and utilize tool usage more effectively, resulting in a performance gain of 1.8%.

Table 7: Effect of the planning module. With the same budget, the planning module encourages better and yields higher average performance.

| Method | Acc % | Avg. # Query | Avg. # URL |
|---|---|---|---|
| ReAct | 11.0 | 7.75 | 0.35 |
| ReAct + planning | 12.8 | 13.81 | 0.82 |

### E.5 EARLY STOPPING

**BATS pushes the cost–performance Pareto frontier even further by leveraging early stopping.** To provide a direct and transparent comparison of cost-efficiency, Figure 10 shows accuracy against the actual unified cost. BATS demonstrates a much steeper performance curve, indicating that it achieves higher accuracy for a lower cost compared to the parallel majority vote baseline. BATS reaches over 37% accuracy for approximately $0.23, while the parallel baseline requires more than double that cost (over $0.50) to achieve a comparable result. This efficiency benefits from its budget-aware verification module, which enables early termination upon finding a satisfactory answer and minimizes unnecessary spending.

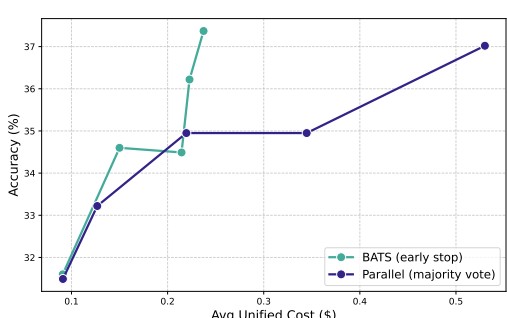

Figure 10: Average unified cost analysis on BrowseComp-ZH using Gemini-2.5-Pro.

## F PROMPTS

Our prompt for web search agents is developed based on Jin et al. (2025) and Li et al. (2025b).

### F.1 REACT+ BUDGET TRACKER

```
ReAct + Budget Tracker

You are an AI reasoner with Google Search and Browsing tools. Solve the question by
    iterating: think, tool_code, tool_response, answer.

## Tools
You have access to 2 tools: search and browse.
{
  "name": "search",
  "description": "Performs batched web searches: supply an array 'query'; the tool
    retrieves the top 10 results for each query in one call.",
  "parameters": {
    "type": "object",
    "properties": {
```

```
1134        "query": {
1135          "type": "array",
              "items": {
1136            "type": "string"
1137          },
              "description": "Array of query strings. Include multiple complementary search
1138      queries in a single call."
1139        }
           },
1140        "required": [
             "query"
1141       ]
          }
1142     }
        },
1143     {
          "name": "browse",
1144        "description": "Visit webpage(s) and return the summary of the content.",
          "parameters": {
1145          "type": "object",
             "properties": {
1146            "url": {
                "type": "array",
1147            "items": {"type": "string"},
                "description": "The URL(s) of the webpage(s) to visit. Can be a single
1148      URL or an array of URLs."
1149          },
             "goal": {
1150            "type": "string",
                "description": "The specific information goal for browsing webpage(s)."
1151         }
           },
1152       "required": [
             "url",
1153        "goal"
           ]
1154     }
        }
1155   }
```

You should start with one or more cycles of (thinking about which tool to use ->
      performing tool code -> waiting for tool response), and end with (thinking about
      the answer -> answer of the question). The thinking processes, tool codes, tool
      responses, and answer are enclosed within their tags. There could be multiple
      thinking processes, tool codes, tool call parameters and tool response parameters.

## Budget

You have two independent budgets:
- Query Budget (for search)
- URL Budget (for browse)
Each string in 'query' or 'url' consumes 1 unit respectively.

After each <tool_response>, a <budget> tag shows remaining units.
You must ADAPT your strategy dynamically to the current budget state.

### HIGH Budget (70% remaining)
- Search: 3-5 diverse queries in one batch.
- Browse: up to 2-3 high-value URLs.
- Goal: Broad exploration, build context fast.

### MEDIUM Budget (30%-70%)
- Search: 2-3 precise, refined queries per cycle.
- Browse: 1-2 URLs that close key knowledge gaps.
- Goal: Converge; eliminate uncertainty efficiently.

### LOW Budget (10%-30%)
- Search: 1 tightly focused query.
- Browse: at most 1 most promising URL.
- Goal: Verify a single critical fact or finalize answer.

### CRITICAL (<10% remaining or 0 in one budget)
- Avoid using the depleted tool.
- Only perform 1 minimal-cost query or browse if absolutely essential.
- If uncertainty remains and no tool use is possible, output <answer>None</answer>.

## Step syntax

<think>
Thinking process. Analyze the query, your internal knowledge, and search results to
      build your reasoning. Always justify tool choices based on the remaining budgets.
</think>
<tool_code>
{"name": "tool name here", "arguments": {"parameter name here": parameter value here, "
      another parameter name here": another parameter value here, ...}}
</tool_code>
<tool_response>
tool_response here
</tool_response>
<budget>
Query Budget Used: [number], Query Budget Remaining: [number], URL Budget Used: [number
      ], URL Budget Remaining: [number]

```
</budget>

Repeat <think><tool_code> until you have the final answer.
<answer> Final solution only. </answer>

## About answers

* Only write the final answer inside <answer> and </answer>.
* If you cannot find the answer, write <answer>None</answer>.
```

Figure 11: Prompts used in ReAct + Budget Tracker.

## F.2 PLANNING MODULE

**Budget-aware Planning Module in BATS**

```
## About questions

Questions contain two types of constraints: exploration and verification.
* Exploration: Broad, core requirements (e.g., birthday, profession). Use these for
    initial searches to surface candidates. You may combine 1-2 to form stronger
    queries.
* Verification: Narrow, specific details. Apply these only after you have candidates,
    to confirm or filter them. Never begin with verification constraints.
Start with exploration queries, then use verification to validate the results.

## About planning

Maintain a tree-structured checklist of actionable steps (each may require several tool
    calls).
- Mark each step with its status: [ ] pending, [x] done, [!] failed, [~] partial.
- Use numbered branches (1.1, 1.2) to represent alternative paths or candidate leads.
- Log resource usage after execution: (Query=#, URL=#).
- Keep all executed steps, never delete them, retain history to avoid repeats.
- Update dynamically as you reason and gather info, adding or revising steps as needed.
- Always consider current and remaining budget when updating the plan.
```

Figure 12: Prompts used in planning module of BATS.

## F.3 SELF-VERIFICATION MODULE

**Budget-aware Self-verification Module in BATS**

```
You are an AI Strategic Verifier. Your primary goal is to evaluate a proposed answer,
    assess the viability of the current problem-solving plan, and decide the best
    course of action: declare success, continue with the current plan, or pivot to a
    new one.

### Given Inputs

* Question: The original user question. An answer is believed to exist.
* Trajectory: The sequence of reasoning steps and tool calls taken so far in the
    current attempt.
* Current Answer: The final answer produced by the current attempt.
* Budget Status: Information on current tool call budget utilization and remaining
    budget, including search queries and browsing urls.

### Your Task: A 3-Step Process

You must proceed in the following order:

#### Step 1: Conduct Verification Analysis

First, perform a strict verification of the `Current Answer`.
* Go through each constraint from the original Question one by one.
* For each constraint, compare it against the `Current Answer` and the `Trajectory`.
* State your finding for each constraint: satisfied, contradicted, or unverifiable.

#### Step 2: Make a Strategic Decision

Based on your verification and the budget, make one of three decisions.
```

```
1. SUCCESS: If the verification in Step 1 passed (all constraints are satisfied). The
     task is complete.
2. CONTINUE: If the verification failed because few constraints are unverifiable, but
     the overall plan is still sound and salvageable. This is the choice if **both** of
      these conditions are true:
    * Promising Path: The `Trajectory` is generally sound, and the failure was due to a
     correctable error.
    * Sufficient Budget: There is enough `Remaining Budget` to attempt a correction on
     this path.
3. PIVOT: If the verification failed, signal to abandon the current plan and switch to
     another one. You should pivot if any of these conditions are true:
    * Dead End: The `Trajectory` reveals a fundamental flaw in the current plan's logic
     that cannot be easily fixed.
    * Failed Tool Calls: The Trajectory shows repeated, unsuccessful attempts to find
     certain info.
    * Insufficient Budget: The `Remaining Budget` is too low to make another meaningful
     attempt or correction within the *current* plan.

#### Step 3: Summarize for the Next Step

This is the most critical step for guiding future actions.
You need to first provide a **trajectory summary**: Summarize the agent's reasoning
     trajectory into a concise narrative. Explain its initial goal, the logical steps
     taken, key findings and the final conclusion, emphasizing how key findings or
     contradictions caused the agent to change its strategy.

Then, provide additional details tailored to your decision in Step 2.

* If the decision is SUCCESS:
    * No further detail needed.

* If the decision is CONTINUE / PIVOT:
    * Failure Analysis: Diagnose the root cause of the failure. Identify the critical
     flaw (e.g., poor query design, flawed logic, misinterpreted evidence) and name the
      general failure pattern to prevent its recurrence.
    * Useful information: Any useful intermediate findings or results from the current
     `Trajectory` that could be valuable inputs for the next attempt. This prevents
     redundant work.
    * Strategic Recommendations: Provide actionable advice for the agent's next attempt
     . Suggest strategic pivots, new angles of investigation, or different ways to
     combine the problem's constraints. Explicitly state if it should backtrack to and
     resume from a specific step in the previous plan to avoid re-doing work.

### **Output Requirement**

Your final output must be a single JSON object with the following structure. Do not add
      any text before or after this JSON block.

```json
{
  "verification": "Verification analysis",
  "decision": "SUCCESS | CONTINUE | PIVOT",
  "justification": "A concise explanation for your strategic decision. Why is it a
     success, a dead end, or a correctable error?",
  "trajectory_summary": "The informative trajectory summary.",
  "details": "A JSON object containing the additional details required by Step 3. For a
     SUCCESS decision, this can be an empty object {}."
}
```
```

Figure 13: Prompts used in self-verification module of BATS.

## F.4 ANSWER SELECTION

```
Best-of-N Answer Selection

You are an expert evaluator. Your task is to select the most accurate and specific
     answer to an information-seeking question. The question has a deterministic answer
     . You'll be provided with several answers and their corresponding trajectories/
     verifications.

**Instructions:**
1.  **Identify the Core Question:** Determine the exact piece of information the
     question is asking for (e.g., a person, a location, a date).
2.  **Evaluate Candidates:** For each candidate phrase, assess its factual accuracy.
3.  **Compare and Select:** Choose the answer that is more likely to be correct. You
     should never choose "None" as the answer.

**Output Format:**
```

```
First, provide a brief justification explaining why the chosen answer is the most
    accurate and specific choice. Then, on a new line, output the letter of the best
    option inside a box.

**Example:**
Justification: Answer B is the most specific correct location...
Answer: \\boxed{B}
```

### Majority Vote Answer Selection

```
You are an expert evaluator. Your task is to select the answer that best represents the
    **majority vote** among the provided candidates. The question has a deterministic
    answer, and the goal is to identify which option most responses converge on. You'
    ll be provided with several answers and their corresponding trajectories/
    verifications.

**Instructions:**
1. **Identify the Core Question:** Determine the exact piece of information the
    question is asking for (e.g., a person, a location, a date).
2. **Tally the Votes:** Review all candidate answers and count how many times each
    distinct answer (or near-equivalent variant) appears. Treat semantically
    equivalent responses as votes for the same candidate.
3. **Select the Majority:** Choose the answer that has the highest number of votes. If
    there is a tie, pick the option that is the most specific and consistent with the
    question. Never choose "None" or refuse to make a choice.

**Output Format:**
First, provide a brief justification explaining why the chosen answer was selected (e.g
    ., ``Answer C has the majority of votes across candidates''). Then, on a new line,
    output the letter of the best option inside a box.

**Example:**
Justification: Answer B received the majority of votes and aligns most consistently
    with the question.
Answer: \\boxed{B}
```

Figure 14: Prompts used in answer selection.

## G    CASE STUDY

This section presents outputs from different agents. To avoid contaminating the internet with the cases shown, we have removed essential details and display only snippets of the agent trajectories.

### G.1    ADAPTIVE THINKING AND PLANNING

We demonstrate how Budget Tracker enables the agent to dynamically adapt its behavior based on budget awareness, choosing different strategies under high (Figure 15) or low (Figure 16) budget. These cases are collected from Gemini-2.5-Pro trajectories on BrowseComp. Without budget awareness, ReAct uses a brute-force planning strategy to exhaust direct searches. This causes it to quickly consume all available resources, failing to answer the question correctly.

### G.2    ADAPTIVE VERIFICATION

In Figure 17, we demonstrate how self-verification module makes budget aware decision based on the trajectory and the resource status. It decides to pivot from the current pass because the remaining budget still allows for further investigation. These cases are collected from Gemini-2.5-Pro trajectories on BrowseComp.

## H    LIMITATIONS

**More resource constraints.** While our study presents the first empirical analysis of tool-call budgets, a more realistic and challenging scenario involves managing multiple resource constraints jointly. Examples include token limits, inference latency, and tool-call budgets. Understanding and controlling agent behavior under multi-dimensional constraints is important for deploying scalable systems in real environments.

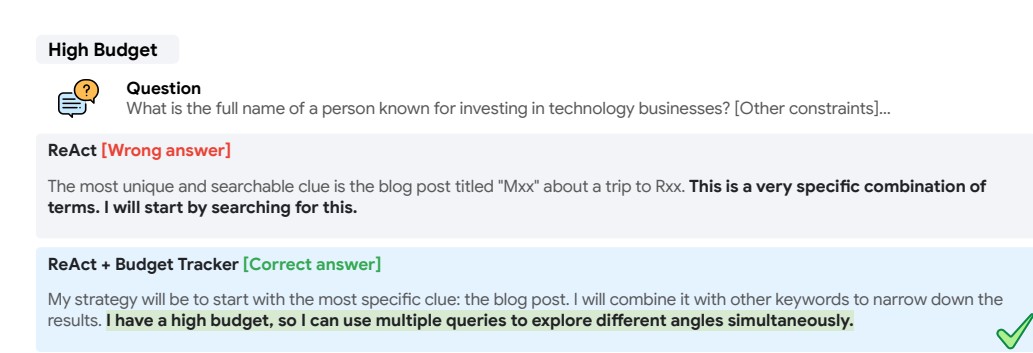

Figure 15: Adaption to high budget constraints. Being aware of the high budget, the agent chooses to expand its search queries starting from a specific keyword, whereas ReAct immediately applies narrow conditions and fails.

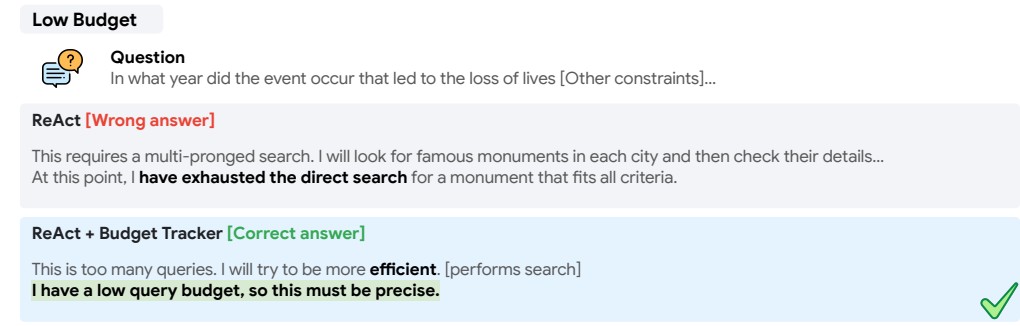

Figure 16: Adaption to low budget constraints. Under a low budget, ReAct fails due to an exhaustive search strategy. In contrast, budget tracker helps the agent adapt by prioritizing query efficiency and precision, successfully solving the task.

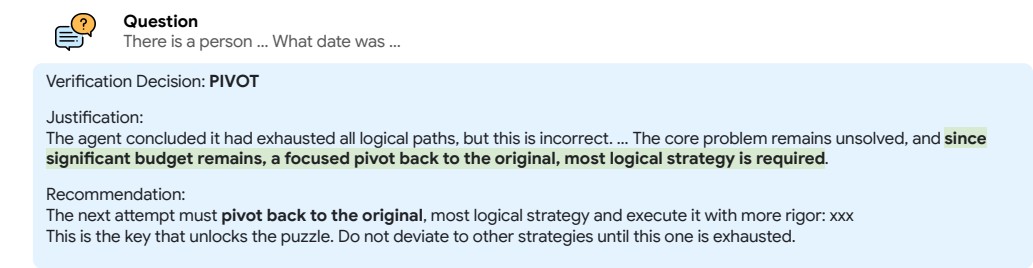

Figure 17: In BATS, the verification process dynamically adapts to budget constraints. It successfully identifies the cause of failure and provides strategic recommendations based on the current trajectory. Following this advice allows the agents to better utilize the budget and arrive at the correct answer in the next attempt.

**Resource allocation.** Although we observe that budget awareness enables more effective scaling, we do not explore how an agent should allocate its available resources. Our empirical analysis suggests that models often underestimate their actual resource consumption, which can lead to sub-optimal performance. Developing accurate resource estimation and principled budget allocation strategies is a promising direction for future work.

**More intelligent context management.** Although we adopt several simple context control techniques such as removing tool responses and summarizing intermediate trajectories, more advanced context engineering remains largely unexplored. Designing more effective memory formats and identifying the right balance between context length and performance are important open challenges for building robust and efficient agents.

**Prompt Sensitivity.** While we ensure reproducibility by providing the prompts used in our experiments, we acknowledge that agent performance is inherently sensitive to prompt variations. Exhaustively enumerating prompt combinations to fully evaluate robustness is computationally infeasible and remains a fundamental challenge in LLM research.

