# OpenReview forum: "Cost-effective Agent Test-time Scaling via Budget-Aware Thinking"
_ICLR.cc/2026/Conference — Submitted to ICLR 2026_

### Official Review · Reviewer_z9K3 · 2025-10-29

**Soundness:** 2
**Presentation:** 2
**Contribution:** 3
**Rating:** 4
**Confidence:** 4

**Summary:**

The paper introduces CATS, a budget-aware system that uses a lightweight tracker to monitor token and tool-call usage, enabling the agent to adapt its planning and verification dynamically based on remaining resources. Through this design, CATS achieves better performance and cost-efficiency by unifying computation and tool-call expenses into a single cost metric. Experimental results on search-intensive benchmarks demonstrate that CATS yields higher accuracy and more favorable scaling curves compared to baseline methods.

**Strengths:**

- The idea of investigating into the cost-performance balance in agent behavior is very interesting and important, especially for real users, as we do not have unlimited budget. Therefore the paper addresses a very important problem in empirical agent usage.

**Weaknesses:**

- The method part writing is a bit unclear as there’s no example provided to illustrate the concept, or prompt shown to the audience how the things are concretely done. Only giving high-level concepts are far from enough. For example, how you categorize clues in two type, what they refer to in the real question, etc.

- The context of using tool call or token can be put into a broader context of whether to use LLM’s internal knowledge (manifested through token output) / external world knowledge (manifested through tool call) to answer a certain question. The decision making between these two aspects is the core of agent behavior. Many related works actually can be discussed from this perspective.

- The results are only performed on Gemini models which may lacks insights, especially when the framework purely rely on prompting, as different model family may react differently towards the same prompt, and the model behavior may also be very sensitive to the prompt content.

- The settings, especially for the baseline, seems very arbitrary and no detailed baseline settings are provided in the main text (e.g. why it’s valid to append the “wait” sentence, what are the alternatively, will it causes differences, and why you do like this). This is important as currently it’s not guaranteed the baselines are compatible, but risk cherry picking which may yield the final conclusion invalid.

**Questions:**

- It’s a smart idea to directly bridge the cost physically through how much money is spent. I am wondering in your setting one tool call equals to approximately how many token input/output price? Will this ratio affect the agent’s behavior / decision making during its planning?

- For the baseline parallel, I am not sure whether the results are comparable. If you set a very large parallel number for the baseline, it definitely needs more budget in total, and since they are parallel run, they cannot re-use the information as effectively as your method. So how you ensure the comparison is fair, especially when the conclusions are about efficiency?

- More insightful analysis can be done on what is causing the improvement in your result. For example, in your analysis, is the raise in tool use the direct cause of you accuracy improvement, or it’s because of the “sequential scaling” (planning is a kind of special “sequential scaling”) that makes model paying more attention to the problem itself?

- The paper will further benefit from doing error analysis including what. Are the common error patterns for failed cases, tool used up, budget reach limit, lack of model capability, or something else? What’s the statistics and how can we potentially improve them? These will guide future research directions more effectively than just informing people the results.

---

> ### Author Response · Authors · 2025-12-04
>
> Thank you for the constructive feedback and insightful questions. We address the concerns and suggestions below.
>
> **W1. Method part is not clear**
>
> Thank you for the helpful feedback. We have substantially revised the method section to improve clarity and completeness. The updated version now includes additional technical details for each module, concrete examples (in case study), and the full prompts used in our framework. We also added a specific section on the budget tracker to explicitly show the impact of budget awareness.
>
>
> **W2. more related work on internal and external knowledge**
>
> Thank you for raising this point. We agree that the distinction between internal knowledge use and external tool use is an important and fundamental question in search agents. We will update the related work section to discuss this broader topic.
>
> **W3. results are only performed on Gemini models**
>
> To address the concern of model dependence, we have extended our experiments to include Claude-Sonnet-4 and revised the manuscript accordingly. We observed that our method consistently outperforms standard baselines on this new model family as well. As shown in the table below, under the same budget of 100 per tool, BATS achieves significant improvements over the baseline across all datasets.
> These results confirm that the benefits of our framework are robust and transferable to other model families as well.
>
> | Method            | Training | BrowseComp | BrowseComp-ZH | HLE-Search |
> |-------------------|----------|------------|----------------|------------|
> | Claude-Sonnet-4 — ReAct       | x        | 12.2       | 29.1          | 20.5       |
> | Claude-Sonnet-4 — ReAct+Budget Tracker     | x        | 14.0       | 31.1          | 23.0       |
> | **Claude-Sonnet-4 — BATS (Ours)** | x        | **19.1**   | **41.5**      | **29.0**   |
>
>
> **W4. Lack of details on baselines**
> > why it’s valid to append the “wait” sentence
>
> We have added additional explanation of the baselines in Appendix B. Appending a “wait” instruction is a simple but widely effective strategy in reasoning settings: prior work [1,2] has shown that such prompts consistently trigger models to perform self-reflection behaviors. As an alternative, agents could explicitly invoke a self-verification step without “wait”; we now include this variant in our ablation results in Appendix C to show that the comparison is fair and the effect is not due to arbitrary prompt choices.
>
>
> > baselines seem arbitrary
>
> We agree that stronger baselines are important and have expanded the results accordingly. In particular, we have extended our comparisons with two recent and strong methods:
>
> (1) Search-o1 [3], a strong baseline that involves scraping web content for every search result.
>
> (2) SLIM [4], a very recent method that employs periodic summarization to maintain a concise context and reduce cost.
>
>
> | Claude-Sonnet-4 | Tool budget | Success Rate | # Tool call |
> | :-------------- | :---------- | :----------- | :---------- |
> | Search-o1 | 50 | 3.7 | 44.1 |
> | Search-o1 | 100 | 7.0 | 79.5 |
> | SLIM | 50 | 10.0 | 27.1 |
> | SLIM | 100 | 10.7 | 28.1 |
> | ReAct + Budget Tracker | 50 | 11.1 | 31.1 |
> | ReAct + Budget Tracker | 100 | 14.0 | 63.5 |
>
> Under the same base model Claude–Sonnet-4 and an equivalent tool budget (50/100), our method with budget tracker achieves a higher accuracy of **14.0%**, outperforming SLIM's **10.7%**. Furthermore, our framework doubles the accuracy over Search-o1 (14.0% vs. 7.0%) while utilizing fewer tool calls (63.5 vs. 79.5).
>
> We also observe that without budget awareness, the most competitive baseline, SLIM, **fails to utilize a larger tool budget**, with its performance converging at 10.7% even when given a 150 tool call budget. In contrast, our framework BATS achieves 19.0% under a budget of 200, providing a significant improvement in scaling performance. We will continue to incorporate additional baseline results across all datasets in the future revision to ensure comprehensive and fair comparisons.

---

> > ### Author Response · Authors · 2025-12-04
> >
> > **Q1. One tool call equals to approximately how many token input/output price**
> >
> > Thank you for your insightful question. In our setting, we estimate the cost of a single tool call to be approximately $0.001, which is derived from an average post-hoc statistical analysis of our experimental trajectories. For different models, it corresponds to roughly 800 input tokens (or ~100 output tokens) for Gemini-2.5-Pro, and roughly 3.3k input tokens (or ~400 output tokens) for Gemini-2.5-Flash. However, the mapping is not perfectly decoupled, since each tool call response also contributes to token consumption in the context.
> >
> > In implementation, our agent does not explicitly optimize for the actual price. It only reasons over the tool-call budget provided by the environment. The actual cost serves as an external evaluation metric rather than an internal reward signal. We consider unifying these resources into a comprehensive "cost-aware" reward model as a promising direction for future work.
> >
> > **Q2. Parallel baseline**
> >
> > To ensure fairness, we perform the comparison based on the total tool-call budget rather than a fixed parallel number. Since different queries require varying number of tool calls, we allow the actual number of parallel sequences to vary dynamically so that both BATS and the Parallel baseline operate under the same budget constraints.
> >
> > Additionally, in section 3.4, we perform a more comprehensive analysis on the effectiveness of budget tracker, showing that our approach can offer better performance and cost-performance scaling curve under both sequential and parallel scaling.
> >
> > **Q3. More insightful analysis**
> >
> > To better understand where the gain comes from, we conduct an ablation analysis using Gemini-2.5-Pro. Across all settings, the agent runs with a fixed budget of 100 tool uses. We compare the full BATS against variants where the planning module, the verification module, or both are removed. We focus on BATS under the early-stopping mechanism, where generation terminates once an answer is verified as successful.
> >
> > | Method                      | BrowseComp | BrowseComp-ZH | HLE-Search |
> > |-----------------------------|------------|----------------|------------|
> > | **BATS**                    | **18.7**   | **39.1**       | **23.0**   |
> > | w/o Planning                | 17.0       | 34.6           | 20.0       |
> > | w/o Verification            | 15.4       | 37.7           | 22.0       |
> > | w/o Planning & Verification | 14.6       | 32.9           | 21.5       |
> > | w/o Planning & Verification & Budget Tracker | 12.6       | 31.5           | 20.5       |
> >
> > The results confirm that both modules offer distinct performance gains. On BrowseComp, the verification module plays a larger role (removing it causes a 3.3% drop), whereas on BrowseComp-ZH, the planning module is more critical (removing it causes a 4.5% drop). The lowest performance occurs when both are removed, showing that the modules work synergistically.
> >
> >
> > In Appendix G, we provide case studies under varying budget constraints. These show that the improvement comes from strategic resource allocation: the agent prioritizes critical search steps under low budgets and expands the search space when higher budgets are available. Together with the ablation, this confirms that the gains result from the agent's budget-aware reasoning rather than indiscriminate tool use.
> >
> >
> >
> > **Q4:Error analysis**
> >
> > We appreciate this suggestion and agree that a structured error analysis provides valuable insight. We conducted a manual inspection of failed cases and identified a unique pattern of overconfidence. This particular failure mode is not budget exhaustion, but the model verifying an incorrect answer as true. We think this "false positive" behavior is linked to model calibration and is difficult to measure via heuristics. We will include an analysis of these error categories in the revision.
> >
> >
> > We hope these clarifications help address your concerns. Thanks again for your time and insightful suggestions.
> >
> > References
> >
> > [1] s1: simple test time scaling
> >
> > [2] A Simple" Try Again" Can Elicit Multi-Turn LLM Reasoning
> >
> > [3] Search-o1: Agentic Search-Enhanced Large Reasoning Models
> >
> > [4] Lost in the Maze: Overcoming Context Limitations in Long-Horizon Information-Seeking

---

### Official Review · Reviewer_r4ZX · 2025-10-30

**Soundness:** 2
**Presentation:** 2
**Contribution:** 3
**Rating:** 2
**Confidence:** 5

**Summary:**

This paper shifts the focus of test-time scaling (TTS) from *token-related scaling* to *tool-call-related scaling*, thereby extending the scope of TTS. It introduces a new problem: how to effectively utilize additional computational budgets allocated for tool calls in tool-augmented agents. To address this, the authors design a *cost-effective agent test-time scaling framework* that scales computation for tool calling with budget-awareness. Using *web search* as a representative task, the proposed scaling method is shown to outperform the base model, several existing trained agents, and some token-based TTS methods.

**Strengths:**

1. The paper presents a novel research problem — *test-time scaling under a tool-call budget* — which fills an important gap in current TTS research and is of high research value.
2. The proposed budget-aware planning and balanced budget usage are thoughtful and practically meaningful design considerations.
3. The problem formulation and motivation are clearly written, especially in the early sections of the paper.

**Weaknesses:**

1. The writing quality fluctuates significantly. The first two sections are well-organized and easy to follow, but the later sections, particularly Section 3, lack critical technical details. While the introduction mentions key challenges such as balancing different budget usage across tools and budget-aware exploration, the method section does not clearly explain how these challenges are addressed. Furthermore, several essential details are missing, which hinder reproducibility — for example, how the agent selects tools, how subsequent planning steps utilize previous tool outputs, and how the agent balances between breadth and depth of exploration. The paper would benefit from additional equations and figures to improve clarity. Finally, the early termination mechanism mentioned in line 398 lacks a corresponding description in the method section, and line 472 contains a capitalization issue.
2. The unified metric introduced in Equation (2) lacks generalization. The definition of *economic cost* does not apply uniformly to both local model usage and local tool functions because the models and tools in local are free.
3. The problem setup, which assumes that each tool call has an explicitly defined budget, is unrealistic. In practice, users typically specify a *total* computational budget, and the allocation across tools should be determined by the proposed method. Moreover, the budget definition should jointly consider both *token budget* and *tool-call budget* to align with the broader definition of test-time scaling.
4. The experiments are insufficient.
    (1) The main experiment should focus on demonstrating how the proposed method improves the *scaling cost–performance curve*, rather than only comparing with base or parallel models. Figure 2’s comparison is therefore inadequate.
    (2) The choice of TTS baselines is limited. The token-based baselines do not include current state-of-the-art methods, and the paper does not explore tool-call-related scaling methods, such as simple BoN or self-refinement on tool calls.
    (3) Evaluating on only one type of task is insufficient to demonstrate the method's generalization.
    (4) The ablation studies in Section 5.3 are not well-designed. The ablation of the planning and budget-tracker modules should be conducted within the proposed framework itself, not on another existing method.

**Questions:**

1. Has the paper considered the impact of different temperature parameters on agent exploration behavior?
2. How is the *parallel* baseline in Figure 2 related to tool-call-based scaling?
3. Could the authors provide additional results for larger budget settings in Figure 3 to show the trend of performance improvement with increased budget?

---

> ### Author Response · Authors · 2025-12-04
>
> Thank you for the constructive feedback and insightful questions. We address the concerns and suggestions below.
>
> **W1. Writing quality**
>
> > lack critical technical details
>
> We have substantially revised the method section to improve clarity and completeness. The updated version now includes additional technical details for each module, concrete examples (in case study), and the full prompts used in our framework (Appendix F). We also added a specific section on the budget tracker to explicitly show the impact of budget awareness (Section 3).
>
> > several essential details are missing
>
> We have added the full prompt in Appendix F to help readers better understand our framework. Our tool-use follows the ReAct paradigm, where the tool output is appended to the context and then fed into subsequent iterations. To balance depth and breadth in exploration, the planning module determines the next actions, while the verification module makes budget-aware decisions on whether to *continue* digging deeper or *pivot* to explore alternative directions. We have also clarified the early-stopping mechanism and corrected several typos in the revised manuscript.
>
> **W2. For unified cost, local models and tool calls are free**
>
>
> Equation (2) is a specific instantiation tailored for the search agents. It serves as a methodological reference for balancing performance and resource consumption. While local models and tools do not incur per-call API fees, they are far from "free" in a systems context. They consume significant computational resources, introduce inference latency, and occupy context window capacity (tokens used for tool call and outputs). We’ll further clarify the generalizability in the revision.
>
>
> **W3. Problem setup is unrealistic**
>
> In production-grade systems (e.g., commercial Deep Research services), hard limits on tool usage are standard safeguards to **prevent infinite loops and uncontrolled costs**. Our work addresses this specific, real-world constraint, filling a gap often overlooked by pure token-based scaling.
>
> We view the optimization of tool-call constrained budgets as a foundational step toward broader test-time scaling. Before achieving a fully unified resource budget (tokens, tools, time, etc.), it is essential to understand how agents behave under hard tool constraints. Our work isolates and addresses this specific challenge. Integrating this into a joint all resource budget is a complex yet promising optimization problem that we leave for future work.

---

> ### Author Response · Authors · 2025-12-04
>
> **W4. Insufficient experiments**
>
> >  More baselines
>
> We agree that stronger baselines are important and have expanded the results accordingly. In particular, we have extended our comparisons with two recent and strong methods:
>
> (1) Search-o1 [1], a strong baseline that involves scraping web content for every search result.
>
> (2) SLIM [2], a very recent method that employs periodic summarization to maintain a concise context and reduce cost.
>
>
> | Claude-Sonnet-4 | Tool budget | Success Rate | # Tool call |
> | :-------------- | :---------- | :----------- | :---------- |
> | Search-o1 | 50 | 3.7 | 44.1 |
> | Search-o1 | 100 | 7.0 | 79.5 |
> | SLIM | 50 | 10.0 | 27.1 |
> | SLIM | 100 | 10.7 | 28.1 |
> | ReAct + Budget Tracker | 50 | 11.1 | 31.1 |
> | ReAct + Budget Tracker | 100 | 14.0 | 63.5 |
>
> Under the same base model Claude–Sonnet-4 and an equivalent tool budget (50/100), our method with budget tracker achieves a higher accuracy of **14.0%**, outperforming SLIM's **10.7%**. Furthermore, our framework doubles the accuracy over Search-o1 (14.0% vs. 7.0%) while utilizing fewer tool calls (63.5 vs. 79.5).
>
> We also observe that without budget awareness, the most competitive baseline, SLIM, **fails to utilize a larger tool budget**, with its performance converging at 10.7% even when given a 150 tool call budget. In contrast, our framework BATS achieves 19.0% under a budget of 200, providing a significant improvement in scaling performance. We will continue to incorporate additional baseline results across all datasets in the future revision to ensure comprehensive and fair comparisons.
>
>
>
> > (2) TTS and BoN methods
>
> We have incorporated comparisons with different scaling strategies (Best-of-N, Majority Vote) and pass@N metrics in Section 3.4 and Appendix E2. The analysis confirms that integrating our budget tracker provides consistent performance gains under various scaling settings. We are continuing to expand the experiments for BATS and will update the manuscript with these additional results.
>
> > (3) More tasks
>
> We appreciate the insightful suggestion. While our primary focus in this work remains on search agents as defined in our problem formulation, we fully agree with the importance of broader applicability. To demonstrate the potential of our framework on complex tasks, we have expanded our evaluation to the Humanity's Last Exam (HLE) in Table 3. Unlike standard web search benchmarks, HLE requires graduate-level reasoning in specialized domains, closely mirroring the cognitive demands of scientific discovery. We are continuing to explore more on other agentic tasks to further extend our contribution.
>
> > (4) Ablations
>
> To isolate the specific contributions of the planning and verification modules, we conducted an ablation analysis using Gemini-2.5-Pro. Across all settings, the agent runs with a fixed budget of 100 tool uses. We compare the full BATS model against variants where the planning module, the verification module, or both are removed. We focus on BATS under the early-stopping mechanism, where generation terminates once an answer is verified as successful.
>
> | Method                      | BrowseComp | BrowseComp-ZH | HLE-Search |
> |-----------------------------|------------|----------------|------------|
> | **BATS**                    | **18.7**   | **39.1**       | **23.0**   |
> | w/o Planning                | 17.0       | 34.6           | 20.0       |
> | w/o Verification            | 15.4       | 37.7           | 22.0       |
> | w/o Planning & Verification | 14.6       | 32.9           | 21.5       |
> | w/o Planning & Verification & Budget Tracker | 12.6       | 31.5           | 20.5       |
>
> The results confirm that both modules offer distinct performance gains. On BrowseComp, the verification module plays a larger role (removing it causes a 3.3% drop), whereas on BrowseComp-ZH, the planning module is more critical (removing it causes a 4.5% drop). The lowest performance occurs when both are removed, showing that the modules work synergistically. We have included detailed analysis in the revised version.

---

> ### Author Response · Authors · 2025-12-04
>
> **Q1. Different temperature**
>
> We follow the common practice of using temperature = 0.7 for agent tasks. For evaluation, we use temperature = 0.0 for answer selection to ensure deterministic scoring.
> To assess its impact, we conduct an additional experiment varying the temperature on a 200-sample subset of BrowseComp using Gemini-2.5-Flash. As shown below, the performance is relatively stable across settings, and temperature = 0.7 yields the best accuracy:
>
> | Temperature  | Acc (%)  |
> | --- | ---- |
> | 0.0 | 15 |
> | 0.3 | 14  |
> | 0.7 | 16   |
> | 1.0 | 14   |
>
>
>
> **Q2. How is parallel scaling related to tool call scaling**
>
> Parallel sampling generates $N$ independent sequences, each executing its own set of tool calls. Therefore, increasing the sample size $N$ directly scales the total tool consumption. We use this as a baseline to demonstrate whether our method utilizes the same tool-call budget more efficiently than parallel scaling with majority vote.
>
> **Q3. Larger budget**
>
> Following your suggestion, we increase the tool call budget to 200 for each tool and test it on a 200 subset of BrowseComp due to resource constraints. Performance improved from 28.0% (Budget=100) to 32.0% (Budget=200), confirming that our framework effectively converts additional compute into better performance.
>
>
>
> | Tool budget | BrowseComp-200 |
> |-------------|----------------|
> | 100         | 28.0           |
> | 200         | 32.0           |
>
> We hope these clarifications help address your concerns. Thanks again for your time and insightful suggestions.
>
> [1] Search-o1: Agentic Search-Enhanced Large Reasoning Models
>
> [2] Lost in the Maze: Overcoming Context Limitations in Long-Horizon Information-Seeking

---

### Official Review · Reviewer_B1Ff · 2025-10-31

**Soundness:** 3
**Presentation:** 2
**Contribution:** 2
**Rating:** 4
**Confidence:** 3

**Summary:**

This paper addresses the challenge of cost-effective test-time scaling for tool-augmented large language model (LLM) agents . The authors distinguish scaling "thinking" (token generation) from "acting" (tool calls) and focus on managing performance under explicit tool-call budgets. They propose CATS (Cost-effective Agent Test-time Scaling), a framework featuring a lightweight budget tracker. This tracker provides a continuous signal of remaining resources to budget-aware planning and budget-aware verification modules. The paper formalizes the budget-constrained optimization problem (Equation 1) and introduces a unified cost metric to account for both token and tool-call costs (Equation 2). Experiments on web search benchmarks show that CATS produces more favorable scaling curves, achieving higher accuracy with fewer tool calls and lower overall unified cost compared to baseline scaling methods.

**Strengths:**

1. The paper is well-organized and uses easy-to-understand language.

2. Clear and relevant problem formulation.

3. Sound and intuitive framework design.

**Weaknesses:**

1. The entire framework seems to rely on manual framework engineering and prompt design, raising concerns about its scalability and generalizability.

2. Model Dependence: Experiments are conducted exclusively on the Gemini series of models.

3. Task Dependence: The framework is instantiated and evaluated exclusively on web search tasks. The paper claims broad applicability to tool-augmented agents , but provides no discussion or evidence for other domains (e.g., software engineering, scientific discovery) where tool-call costs and task structures are vastly different.

4. Prompt Sensitivity: The framework's effectiveness is likely sensitive to the quality of the prompts used to guide the planning (Sec 3.2) and verification (Sec 3.3)  modules. The paper does not discuss this potential sensitivity. No direct evidence found in the manuscript.

5. Incomplete ablation: It remains unclear how much of the performance gain from the full CATS model (Table 1) comes from the planning module (Sec. 3.2) versus the verification/early-stopping logic (Sec. 3.3), as these components are not tested in isolation from each other.

**Questions:**

1. Can the authors provide results on a wider range of models, such as the GPT series or open-source models?

2. Can the framework be tested on additional tasks beyond web search?

3. Can the authors provide more comprehensive ablation studies?

---

> ### Author Response · Authors · 2025-12-04
>
> Thank you for the constructive feedback and insightful questions. We address the concerns and suggestions below.
>
>
> **W1&W4. Rely on manual framework engineering and prompt design**
>
> We agree that prompt sensitivity is an important consideration for any LLM-based framework. We have released all prompts in Appendix F to ensure full transparency and reproducibility.
> To empirically address the concerns regarding scalability and robustness, we have **expanded our experiments to include a different model family (Claude-Sonnet-4) and a distinct dataset (HLE)**. BATS demonstrated consistent performance across these new settings, indicating that our framework design is robust and not over-fitted to a specific model or task.
> Finally, we have updated the Limitations section to explicitly discuss the potential concerns of prompt sensitivity.
>
> **W2&Q1. Experiments only conducted on Gemini models**
>
> To address the concern of model dependence, we have extended our experiments to include Claude-Sonnet-4 and revised the manuscript accordingly. We observed that our method consistently outperforms standard baselines on this new model family as well. As shown in the table below, under the same budget of 100 per tool, BATS achieves improvements over the baseline across all datasets.
> These results confirm that the benefits of our framework are robust and transferable to other model families as well.
>
> | Method            | Training | BrowseComp | BrowseComp-ZH | HLE-Search |
> |-------------------|----------|------------|----------------|------------|
> | Claude-Sonnet-4 — ReAct       | x        | 12.2       | 29.1          | 20.5       |
> | Claude-Sonnet-4 — ReAct+Budget Tracker     | x        | 14.0       | 31.1          | 23.0       |
> | **Claude-Sonnet-4 — BATS (Ours)** | x        | **19.1**   | **41.5**      | **29.0**   |
>
> **W3&Q2. Additional tasks**
>
> We appreciate the insightful suggestion. While our primary focus in this work remains on search agents as defined in our problem formulation, we fully agree with the importance of broader applicability. To demonstrate the potential of our framework on complex tasks, we have expanded our evaluation to the Humanity's Last Exam (HLE) in Table 3. Unlike standard web search benchmarks, HLE requires graduate-level reasoning in specialized domains, closely mirroring the cognitive demands of scientific discovery. We are continuing to explore more on other agentic tasks to further extend our contribution.
>
>
> **W5&Q3. Ablations on planning and verification**
>
> To isolate the specific contributions of the planning and verification modules, we conducted an ablation analysis using Gemini-2.5-Pro. Across all settings, the agent runs with a fixed budget of 100 tool uses. We compare the full BATS model against variants where the planning module, the verification module, or both are removed. We focus on BATS under the early-stopping mechanism, where generation terminates once an answer is verified as successful.
>
> | Method                      | BrowseComp | BrowseComp-ZH | HLE-Search |
> |-----------------------------|------------|----------------|------------|
> | **BATS**                    | **18.7**   | **39.1**       | **23.0**   |
> | w/o Planning                | 17.0       | 34.6           | 20.0       |
> | w/o Verification            | 15.4       | 37.7           | 22.0       |
> | w/o Planning & Verification | 14.6       | 32.9           | 21.5       |
>
>
> The results confirm that both modules offer distinct performance gains. On BrowseComp, the verification module plays a larger role (removing it causes a 3.3% drop), whereas on BrowseComp-ZH, the planning module is more critical (removing it causes a 4.5% drop). The lowest performance occurs when both are removed, showing that the modules work synergistically. We have included detailed analysis in the revised version.
>
> We hope these clarifications help address your concerns. Thanks again for your time and insightful suggestions.

---

### Official Review · Reviewer_AUaU · 2025-11-01

**Soundness:** 2
**Presentation:** 3
**Contribution:** 2
**Rating:** 4
**Confidence:** 4

**Summary:**

This paper addresses cost-effective test-time scaling for tool-augmented agents by tracking and enforcing budgets not only as a function of thinking tokens but also---importantly---as a function of tool calls, which are both unified by means of their monetary costs. The authors propose CATS (Cost-effective Agent Test-time Scaling), which performs "constraint decomposition" and proceeds to either "explore" or "verify" each branch from a question decomposition, while following a global budget constraint that governs how much exploration and verification are allowed. Experiments on BrowseComp and BrowseComp-zh benchmarks with Gemini Flash or Pro, covering relatively naive sampling-based baselines, shows CATS outperforming the naive baselines, the pretrained models alone, and most custom trained models (although differences in backbone models limit all comparisons but the ones against the naive baselines). Authors conclude with sensible analyses about early stopping even with remaining budget and scaling behavior vs. naive baseline.

**Strengths:**

1. Very practically relevant topic.

2. The premises are intuitive and easy to understand---the idea of unifying two natures of costs by their monetary values, which then allows for a global budget to be tracked and enforced, makes a lot of sense.

3. The paper is mostly clear and straightforward.

**Weaknesses:**

1. Authors show gains over naive baselines that do not attempt to score and prioritize branches in the tree search space in any way. There is a limited analysis of "ReAct + planning," which still does not reflect the state-of-the-art in planning. Importantly, the absence of strong baselines that estimate branch "goodness" without explicitly tracking monetary costs makes it hard to determine if CATS truly is more cost-effective than existing methods.
In this sense, authors should consider running stronger baselines like "Reasoning with Language Model is Planning with World Model" (EMNLP 2023), that applies MCTS with goodness estimates; "ToolChain*: Efficient Action Space Navigation in Large Language Models with A* Search" (ICLR 2024), that applies A* with goodness and non-monetary cost estimates; or "Q*: Improving Multi-step Reasoning for LLMs with Deliberative Planning," that applies MCTS with different goodness estimates; and showing to what extent they exceed monetary cost constraints that CATS is otherwise able to enforce. If true, claims would have much better support than with current comparisons.

2. Why do authors claim that "tool-call budget" is "more relevant" than "token-based budget" (lines 111-112)? If one of the contributions is to unify the cost metric by means of monetary costs, then it would be natural to substantiate this statement with an analysis of the real dollar amounts associated to each resource in the experiments performed. Additionally, to call one budget "more relevant" goes against the motivation of unifying the two---maybe stating that "tool-call budget" is highly relevant but underexplored would be more appropriate.

**Questions:**

As described in the weaknesses, a question:

1. Why do authors claim that "tool-call budget" is "more relevant" than "token-based budget" (lines 111-112)? If one of the contributions is to unify the cost metric by means of monetary costs, then it would be natural to substantiate this statement with an analysis of the real dollar amounts associated to each resource in the experiments performed. Additionally, to call one budget "more relevant" goes against the motivation of unifying the two---maybe stating that "tool-call budget" is highly relevant but underexplored would be more appropriate.

Suggestion:

1. On lines 254-256, consider at least adding a citation that supports the statement that summarization makes exploration "better informed." Otherwise, ideally, this could have been ablated in the Appendix to better understand the proposed methods.

---

> ### Author Response · Authors · 2025-12-04
>
> We thank the reviewer for the constructive feedback and for recognizing the practical relevance, reasonable design and clarity. We address the concerns and suggestions below.
>
> **W1. more baselines**
>
> We agree that stronger baselines are important and have expanded the results accordingly. While the suggested tree-search methods have not been explored in search agent tasks and typically require broad exploration that leads to substantial computational overhead, we have extended our comparisons with two recent and strong methods:
>
> (1) Search-o1 [1], a strong baseline that involves scraping web content for every search result.
>
> (2) SLIM [2], a very recent method that employs periodic summarization to maintain a concise context and reduce cost.
>
> | Claude-Sonnet-4 | Tool budget | Success Rate | # Tool call |
> | :-------------- | :---------- | :----------- | :---------- |
> | Search-o1 | 50 | 3.7 | 44.1 |
> | Search-o1 | 100 | 7.0 | 79.5 |
> | SLIM | 50 | 10.0 | 27.1 |
> | SLIM | 100 | 10.7 | 28.1 |
> | ReAct + Budget Tracker | 50 | 11.1 | 31.1 |
> | ReAct + Budget Tracker | 100 | 14.0 | 63.5 |
>
> Under the same base model Claude–Sonnet-4 and an equivalent tool budget (50/100), our method with budget tracker achieves a higher accuracy of **14.0%**, outperforming SLIM's **10.7%**. Furthermore, our framework doubles the accuracy over Search-o1 (14.0% vs. 7.0%) while utilizing fewer tool calls (63.5 vs. 79.5).
>
> We also observe that without budget awareness, the most competitive baseline, SLIM, **fails to utilize a larger tool budget**, with its performance converging at 10.7% even when given a 150 tool call budget. In contrast, our framework BATS achieves 19.0% under a budget of 200, providing a significant improvement in scaling performance. We will continue to incorporate additional baseline results across all datasets in the future revision to ensure comprehensive and fair comparisons.
>
>
>
> **W2&Q1. why is tool-call budget more relevant than token budget?**
>
> Thank you for your insightful suggestion. We acknowledge that both tool calls and tokens account for non-negligible costs. We prioritize the tool-call budget over the token budget for two primary reasons:
>
> 1. Performance correlation: As evidenced by [3], the number of environment interactions is a key determinant of agent performance. Agents significantly benefit from more frequent interactions, making this a critical metric to optimize.
>
> 2. Gap in the literature: While token scaling is well-studied, we totally agree that tool call remains a unique and under-explored dimension of agent scaling. To this end, our work aims to bridge this gap by providing a focused analysis of agent scaling under tool-call constraints.
>
> >  an analysis of the real dollar amounts associated to each resource
>
> Regarding the suggestion to report real dollar amounts, we have included a detailed breakdown of the cost associated with each resource type in Appendix E.3.
>
>
> **Suggestion 1**
> Thank you for the suggestion! We have included a very recent work [4] as reference, which shows that summarization helps reduce noisy information and maintain an effective exploration trajectory.
>
>
> We hope these clarifications help address your concerns. Thanks again for your time and insightful suggestions.
>
> [1] Search-o1: Agentic Search-Enhanced Large Reasoning Models
>
> [2] Lost in the Maze: Overcoming Context Limitations in Long-Horizon Information-Seeking
>
> [3] MiroThinker: Pushing the Performance Boundaries of Open-Source Research Agents via Model, Context, and Interactive Scaling
>
> [4] Lost in the Maze: Overcoming Context Limitations in Long-Horizon Agentic Search

---

### Author Response · Authors · 2025-12-04
**General Response**

We sincerely thank all the reviewers for their insightful questions and constructive feedback. We are encouraged by the consensus on our work’s core strengths:

- **Clear motivation and practical relevance** to real-world agent usage (reviewer: AUaU, r4ZX, z9K3)
- **Clear problem formulation** and **well-designed framework** (reviewer: B1Ff, r4ZX)
- **Novel and valuable research direction** that fills an important gap in test-time scaling (reviewer: r4ZX)
- **Sound design of budget and unified cost formulation** (reviewer: AUaU)


We have strengthened the manuscript by carefully addressing the reviewers' suggestions:

* We **clarified and deepened the motivation and contribution for budget-aware test-time scaling**, and introduced a new Section 3 that explains the design principles, analysis, and insights behind our approach. We have also renamed our method from Cost-effective Agent Test-time Scaling (CATS) to Budget-Aware Test-time Scaling (BATS) to emphasize the methodological focus and further strengthen the contribution on budget awareness.
* We have **expanded our experiments** to include more tasks (HLE) and more models (Claude-Sonnet-4). The results consistently confirm that budget awareness and BATS provide significant improvements and demonstrate a better cost-performance scaling curve.
* We have **expanded our analysis**, providing detailed explanations through additional ablation study (Appendix C), comprehensive cost analysis (Appendix E3) and case studies (Appendix G).
* We provided further comparison with **more advanced baselines (Search-o1 and SLIM)** in our response.
* We have also refined the writing in the method section for greater detail and appended all prompts used in Appendix F for full transparency and reproducibility.

Thank you again for your valuable time and effort in reviewing our work.

---

### Meta-Review · Area_Chair_VvKg · 2026-01-18

**Summary:**

This paper studies test-time scaling for tool-augmented agents under explicit tool-call budgets and proposes a budget-aware planning and verification framework (BATS). Reviewers agreed that the problem is practically relevant and that reasoning about budget constraints is interesting. However, the overall assessment in the reviews is negative.

**Reviewer Concerns:**

A primary concern raised in the reviews is limited novelty. Reviewers noted that the approach relies largely on system design choices and prompt engineering to track and manage tool-call budgets, and questioned whether it introduces a principled algorithmic advance beyond intuitive heuristics. While the rebuttal clarifies design motivations and implementation details, these clarifications do not materially change how the contribution is characterized in the reviews.

Reviewers also raised concerns about the problem formulation and generality. The assumption of explicit per-tool budgets was viewed as restrictive or unrealistic in many settings, and the unified cost metric was questioned. Although the rebuttal justifies this formulation, the concern remains central in the reviews.

The experimental evaluation was another major issue. Reviewers pointed out limited task diversity, a strong focus on search agents, and initially narrow baseline coverage. The rebuttal adds additional baselines, models, tasks, and ablations, but these additions do not fully address concerns about generality and strength of evidence raised in the original reviews.

**Reviewer Scores:**

Reviewer AUaU (score: 4) acknowledged the practical relevance of budget-aware agent execution but raised concerns about missing strong planning baselines, unclear claims about tool-call relevance, and limited experimental scope. These concerns relate to the core framing and evaluation, and a substantial score increase would be unlikely.

Reviewer B1Ff (score: 4) viewed the idea as reasonable but emphasized reliance on manual prompt engineering, limited task and model diversity, and insufficient ablations. The reviewer explicitly expressed neutrality toward the final decision, suggesting their assessment would likely remain around the original score.

Reviewer z9K3 (score: 4) found the problem important but raised concerns about clarity, baseline fairness, and generality beyond search agents. These issues are unlikely to be resolved through clarification alone.

Reviewer r4ZX (score: 2) expressed strong concerns about the problem setup, experimental design, baseline selection, and technical clarity, leading to a strong rejection. Given the nature of these criticisms, a meaningful upward revision of the score would be unlikely.

Overall, even accounting for the rebuttal, the expected score changes would not indicate a shift toward a unanimous acceptance.

---

### Decision · Program_Chairs · 2026-01-26

Reject